# Gemcitabine and ATR inhibitors synergize to kill PDAC cells by blocking DNA damage response

Stefanie Höfer [1], Larissa Frasch [1], Sarah Brajkovic[1], Kerstin Putzker [2], Joe Lewis[2], Hendrik Schürmann [3,4,5], Valentina Leone[6], Amirhossein Sakhteman[1], Matthew The [1], Florian P Bayer[1], Julian Müller [1], Firas Hamood[1], Jens T Siveke [3,4], Maximilian Reichert [6,7] & Bernhard Kuster [1,7✉]

## Abstract

**The DNA-damaging agent Gemcitabine (GEM) is a first-line treatment for pancreatic cancer, but chemoresistance is frequently observed. Several clinical trials investigate the efficacy of GEM in combination with targeted drugs, including kinase inhibitors, but the experimental evidence for such rationale is often unclear. Here, we phenotypically screened 13 human pancreatic adenocarcinoma (PDAC) cell lines against GEM in combination with 146 clinical inhibitors and observed strong synergy for the ATR kinase inhibitor Elimusertib in most cell lines. Dose-dependent phosphoproteome profiling of four ATR inhibitors following DNA damage induction by GEM revealed a strong block of the DNA damage response pathway, including phosphorylated pS468 of CHEK1, as the underlying mechanism of drug synergy. The current work provides a strong rationale for why the combination of GEM and ATR inhibition may be useful for the treatment of PDAC patients and constitutes a rich phenotypic and molecular resource for further investigating effective drug combinations.**

**Keywords** Pancreatic Cancer; Kinase Inhibitors; Drug Combination Screening; Phosphoproteomics; DNA Damage Response
**Subject Categories** Cancer; DNA Replication, Recombination & Repair

## Introduction

Pancreatic ductal adenocarcinoma (PDAC) is a devastating disease and, unlike for other malignancies, survival rates have barely increased over the past decades (Jemal et al, 2013; Siegel et al, 2021; Sun et al, 2014). Patients diagnosed with early, localized PDAC typically undergo surgical resection, often complemented by chemotherapy or radiotherapy to enhance disease control. Unfortunately, the majority of pancreatic tumors are diagnosed late,

rendering surgical intervention ineffective and, instead, chemotherapy is the primary systemic treatment modality (Park et al, 2021). For many years, the nucleoside analog Gemcitabine (GEM) has been a frontline drug in PDAC, inducing DNA damage and replication stress response in dividing cells by disrupting DNA synthesis (Burris et al, 1997; Park et al, 2021; Plunkett et al, 1995). In current clinical practice, GEM is often administered in combination with albumin-bound Paclitaxel particles (nab-Paclitaxel) as the combination substantially improves therapeutic efficacy (Kang et al, 2018). Another standard therapy is FOLFIRINOX, a multi-chemodrug regimen including folinic acid, 5-fluorouracil, Irinotecan, and Oxaliplatin (Conroy et al, 2011). While FOLFIRINOX can achieve superior therapeutic outcomes, it also comes with higher toxicity, which limits its use to patients with good overall performance status (Conroy et al, 2011; Le et al, 2016). A major unsolved clinical issue is that patients develop chemoresistance, leading to disease progression (Zeng et al, 2019).

These clinical challenges demand innovative therapies. One conceptually promising strategy is the combination of chemotherapy and targeted drugs such as kinase inhibitors (Garcia-Sampedro et al, 2021; Lei et al, 2019; Lopez and Banerji, 2017). Past advances in the molecular characterization of pancreatic tumors have unveiled key pathways that are critical during disease development and progression (Jones et al, 2008; Sinkala et al, 2020; Waddell et al, 2015). Notably, early pancreatic carcinogenesis is driven by oncogenic mutations in KRAS (> 90%), leading to aberrant MAPK signaling that dysregulates cell proliferation, survival, and differentiation (Eser et al, 2014). Moreover, proteins associated with angiogenesis, insulin signaling, or the AKT/mTOR pathway are frequently overexpressed, possibly also promoting the progression of pancreatic tumors towards more severe phenotypes (Fang et al, 2023). Kinase inhibitors (KI) can effectively inhibit many cancer-related signaling pathways and are important cancer therapeutics today. Currently, at least 30 kinase inhibitors are under clinical investigation for PDAC (phase II or III), either as single agents or in combination with chemotherapy, most commonly GEM (Fang et al, 2023). It is noteworthy that many of these investigated inhibitors are approved drugs in other indications. Such drug

[1]Chair of Proteomics and Bioanalytics, Technical University of Munich, Freising, Germany. [2]Chemical Biology Core Facility, EMBL Heidelberg, Heidelberg, Germany. [3]Bridge Institute of Experimental Tumor Therapy (BIT) and Division of Solid Tumor Translational Oncology (DKTK), West German Cancer Center, University Hospital Essen, University of Duisburg-Essen, Essen, Germany. [4]German Cancer Consortium (DKTK), partner site Essen, a partnership between German Cancer Research Center (DKFZ) and University Hospital Essen, Essen, Germany. [5]Department of Medical Oncology, West German Cancer Center, University Hospital Essen, Essen, Germany. [6]Department of Internal Medicine II, University Hospital Rechts der Isar, Technical University Munich, Munich, Germany. [7]German Cancer Consortium (DKTK), Partner Site Munich, Munich, Germany.
✉E-mail: kuster@tum.de

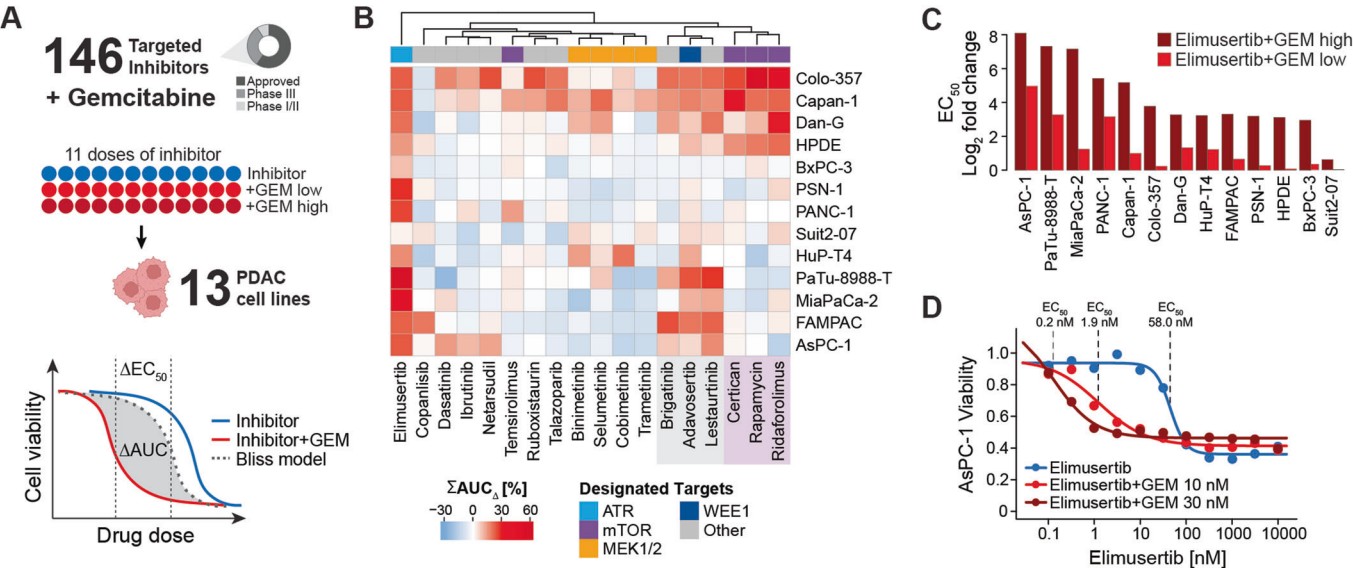

**Figure 1. Drug combination screen reveals ATR inhibitor Elimusertib to synergize with GEM in most PDAC cell lines.**

(A) Drug combination screen of 146 clinically relevant inhibitors and GEM in 13 PDAC cell lines. Eleven doses of inhibitor were tested alone or in combination with two doses of chemodrug (GEM high, GEM low), and synergy was assessed using the Bliss model of independence. (B) Heatmap displaying the summed shift in AUC ($\Sigma AUC_\Delta$, in %) for the 18 drugs that synergized with GEM in at least one cell line. A higher $\Sigma AUC_\Delta$ indicates greater synergy. Drugs are annotated with their designated target proteins. Two clusters of drugs identified from hierarchical clustering are highlighted in gray and purple. (C) Fold change in $EC_{50}$ ($\log_2$) upon combination of ATRi Elimusertib and GEM high (dark red) or GEM low (light red) for all cell lines. (D) AsPC-1 cell line viability after treatment with ATRi Elimusertib alone (blue) or in combination with either 30 nM GEM (dark red) or 10 nM GEM (light red) relative to vehicle. Source data are available online for this figure.

repurposing holds the potential to accelerate clinical development by leveraging well-characterized drug safety profiles and pharmacokinetics (De Lellis et al, 2021).

In PDAC, the only approved repurposed targeted therapy in combination with GEM is Erlotinib, a small molecule EGFR inhibitor initially developed for non-small cell lung cancer (NSCLC) (Kelley and Ko, 2008). The approval came as a result of a phase III trial in which the combination displayed a very modest (two weeks) improvement in terms of overall survival over GEM alone in advanced PDAC patients (Moore et al, 2007). Further post-approval trials showed that the clinical impact of this drug combination was marginal (Abrams et al, 2020; Biagi et al, 2023; Sinn et al, 2017). In NSCLC, the success of Erlotinib was, in part, due to the response association to certain EGFR mutations, but no such molecular subgroup or predictive response marker could be established for PDAC (Tzeng et al, 2007). Moreover, molecular evidence that this drug combination works in a PDAC-specific context remain sparse (Bartholomeusz et al, 2011; Chen et al, 2018; Miyabayashi et al, 2013). We argue that the development of GEM/KI combination therapies in PDAC would immensely benefit from a mechanistic rationale based on specific molecular drug-response biomarkers.

Therefore, in this study, we aimed to discover synergistic combinations of GEM and KI in PDAC and to elucidate the underlying molecular mechanisms of action(s) (MoA) of the observed synergies. First, we screened for drug synergy between GEM and 146 targeted agents (almost all approved and phase III kinase inhibitors to date) in 13 PDAC cell lines. This screen identified the ATR inhibitor Elimusertib to be effective in almost all cell lines, prompting our interest in studying the underlying MoA.

Therefore, we characterized the impact of four investigational ATR inhibitors on cellular signaling using a recently introduced dose-resolved phosphoproteomic approach called decryptM (Zecha et al, 2023). Both phenotypic and molecular datasets are available in ProteomicsDB (Lautenbacher et al, 2022) for further exploration (https://proteomicsdb.org). Analysis of the proteomic data revealed strong DNA damage induction by GEM and simultaneous suppression of DNA repair pathways as the MoA of the drug combination. These results provide a molecular rationale for trying the combination of GEM and ATR inhibitors in PDAC patients.

## Results

### Phenotypic combination screen of Gemcitabine with molecularly targeted cancer drugs

To systematically identify established drugs that possibly synergize with GEM, we performed a phenotypic cell viability screen with 146 targeted agents alone as well as in combination with GEM in 12 human PDAC cell lines and one immortalized pancreatic duct cell line (HPDE; all 13 are referred to as PDAC for simplicity; Fig. 1A, Appendix Table S1). To facilitate potential clinical translation, we focused on clinically advanced compounds (87 approved, 46 phase III and 13 phase I or II; Dataset EV1). Of the 146 drugs, 140 were kinase inhibitors, three PARP1 inhibitors (Olaparib, Talazoparib, Niraparib), and inhibitors of STAT3 (Napabucasin), XPO1 (Selinexor), and SMO (Glasdegib). Cells were treated with 11 doses of the targeted drugs spanning concentrations between 170 pM and 10,000 nM and tested alone or in combination with two fixed

concentrations of GEM (GEM low, GEM high; see below), resulting in more than 1800 pairs of drugs and cell lines. Cell viability was assessed by measuring ATP levels to indicate metabolic activity. A median Z-prime of 0.87 was calculated from both positive and negative controls across the 195 screening plates (384-well format; see Methods for details), indicating high data quality and robustness of the screen.

In the single-agent screen, 50 of 146 compounds showed efficacy in at least one cell line (area under the curve (AUC) < 80%, $-\log_{10}$ effective concentration 50 (pEC$_{50}$) > 6, and goodness of curve fit of squared Pearson correlation coefficient ($R^2$) > 0.8; Dataset EV2). Eighteen inhibitors were efficacious in more than half of the cell lines (Fig. EV1A). The two most active drugs were the cell cycle inhibitors Dinaciclib (CDKi) and Volasertib (PLKi), with median AUCs of 40% and 54% across all cell lines, respectively. Most cell lines also responded to the mTOR inhibitor Sapanisertib (median AUC = 54%). Interestingly, inhibition of mTOR by Rapamycin, Everolimus, Temsirolimus, and Ridaforolimus showed much less efficacy (median AUCs > 80%), suggesting off-targets as the cause of the observed efficacy of Sapanisertib. PDAC cells also responded to inhibition of MEK1 and MEK2 (Cobimetinib, Copanlisib, Trametinib) with median AUCs of < 70%, and to the SRC inhibitor Tirbanibulin (KX2-391) with a median AUC of 60%. GEM alone generally caused stronger responses than the targeted drugs, with AUCs ranging from 21% to 86%, and EC$_{50}$ values of as low as 1 nM (Fig. EV1B, Dataset EV3). Next, we analyzed the drug sensitivity of immortalized duct cell line HPDE to detect effects unrelated to pancreatic cancer. HPDE cells were particularly sensitive towards the CDKi Dinaciclib (AUC = 38%) and also responded to the mTOR inhibitor Sapanisertib, the XPO1 inhibitor Selinexor, and the receptor tyrosine kinase inhibitors (RTKi) Afatinib (EGFR), Neratinib (HER2), and Dacomitinib (EGFR; median AUCs < 60%).

Based on these results, GEM concentrations for the combination screen were set as follows: 10 nM and 30 nM GEM (GEM low, GEM high) for the four least GEM-sensitive cell lines (PANC-1, AsPC-1, MiaPaCa-2, PaTu-8988-T: EC$_{50,\text{GEM}} \gtrsim$ 10 nM), and 1 nM and 3 nM GEM (GEM low, GEM high) for all other cell lines (EC$_{50,\text{GEM}}$ < 10 nM; Dataset EV3). All phenotypic screening data can be interactively visualized and explored in ProteomicsDB (https://proteomicsdb.org/analytics/cellSensitivity).

## Most targeted drugs do not synergize with GEM

Combinations of drugs may lead to stronger or weaker phenotypic effects depending on what biology they target. In addition, positive combined effects can be simply additive or synergistic. To estimate if any drug combination yielded a desired synergistic effect, we applied the Bliss model of independence (Bliss, 1939) (see Methods for details). In essence, the Bliss model predicts the dose–response curve for a drug combination treatment based on the two single-drug dose–response curves under the assumption that effects are additive. If the experimentally determined dose–response curve of the same combination is more potent than the Bliss model, the effects are termed synergistic (Fig. 1A). For the purpose of this study, the magnitude of synergy was quantified by summing up the observed difference in AUC ($\Sigma\text{AUC}_\Delta$) for the two GEM concentrations. Of all 146 drugs, 18 showed synergy with GEM in at least one cell line (using the following thresholds: $\Sigma\text{AUC}_\Delta$ > 10%, and AUC < 80%, pEC$_{50}$ > 6, $R^2$ > 0.8 in at least one of the two GEM combinations; Fig. EV1C, Dataset EV2).

Clustering of $\Sigma\text{AUC}_\Delta$ values grouped drugs with the same designated targets (Fig. 1B). No synergy was observed in any cell line for the potent cell cycle inhibitors Dinaciclib and Volasertib that showed strong effects as single agents (Dataset EV2). Further, we could not detect synergy for GEM and EGFR inhibition even though this drug combination is in clinical use for PDAC patients (Kelley and Ko, 2008). Moreover, none of the other 46 tested RTKi had any synergistic effect when combined with GEM (Fig. EV1C). Although the current panel of cell lines may not be sufficiently representative of all potential PDAC subtypes, this general lack of synergy between RTKi and GEM in vitro is in line with the observation that the combination of GEM and EGFRi Erlotinib is also rather ineffective in the clinic (Abrams et al, 2020; Biagi et al, 2023; Sinn et al, 2017). This highlights the importance of rational combination treatment design.

## mTOR and WEE1/CHEK1 inhibitors synergize with GEM in certain cellular contexts

Among the 18 synergistic drug combinations were 17 kinase inhibitors. The PARP1-inhibitor Talazoparib showed synergy in two cell lines. Two groups of inhibitors commonly sensitized a few but not all PDAC cell lines, indicating context-dependent drug synergy. For instance, the three mTOR inhibitors Ridaforolimus, Rapamycin, and Everolimus synergized with GEM in Colo-357, Capan-1, and Dan-G cells. We note that these drugs also strongly sensitized the immortalized HPDE cells to GEM from which one may expect increased issues with clinical toxicity. Another three kinase inhibitors chemo-sensitized a larger set of PDAC cell lines, namely the WEE1 inhibitor Adavosertib (7 cell lines), the ALK inhibitor Brigatinib, and the broad-spectrum inhibitor Lestaurtinib (5 cell lines each). Albeit the distinct designated targets of these drugs, their synergistic behavior across multiple cell lines could be caused by a common underlying mechanism, such as off-target effects. To identify candidate off-targets, we mined two large-scale KI target profiling datasets based on the Kinobeads technology (Klaeger et al, 2017a; Data ref: Klaeger et al, 2017b; Reinecke et al, 2023a; Data ref: Reinecke et al, 2023b) (Dataset EV4). In both studies, Adavosertib showed high affinity to three kinases (its designated target WEE1, apparent dissociation constant ($K_D^{\text{app}}$) = 12 nM; ADK, $K_D^{\text{app}}$ = 12 nM; MAP3K4, $K_D^{\text{app}}$ = 52 nM). Lestaurtinib and Brigatinib potently interacted with > 20 kinases ($K_D^{\text{app}}$ < 100 nM), including CHEK1 ($K_D^{\text{app}}$ = 70 nM). Despite the lack of one single common target protein, we note that CHEK1 and WEE1 both act as key checkpoint regulators in response to DNA damage (as induced by GEM), suggesting that rather the same biological process explains the observed synergy (O'Connell et al, 1997).

## GEM and ATR inhibitors synergize across almost all PDAC cell lines

While the majority of drugs did not generate synergy in most cell lines, there was one notable exception. The ATR inhibitor (ATRi) Elimusertib showed synergy in 12 of 13 cell lines (Fig. 1B; Dataset EV2). The strong chemosensitizing effects manifested in > 10-fold improved EC$_{50}$ values in nearly all affected cell lines (Fig. 1C). Interestingly, strongest synergy between Elimusertib and GEM was observed in the four cell lines that were the least sensitive to GEM monotherapy. Potency was most enhanced in AsPC-1 cells,

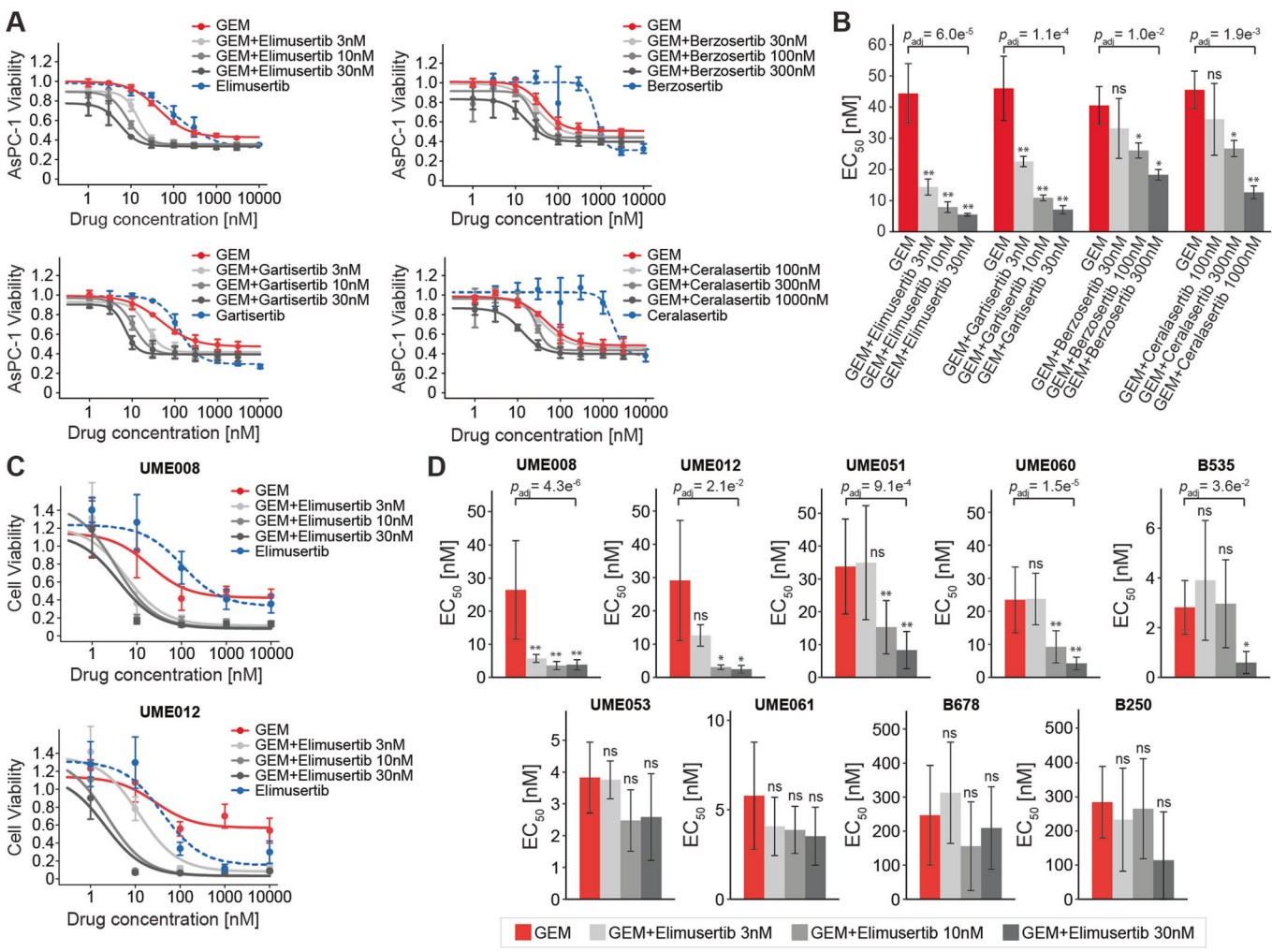

**Figure 2. Synergy of Elimusertib and GEM in PDAC cells can be generalized to other ATR inhibitors.**

(A) AsPC-1 cell line viability after treatment with GEM alone (red), GEM in combination with three sub-EC$_{50}$ doses of ATR inhibitor (shades of gray), or ATR inhibitor alone (blue dotted) relative to vehicle. (B) EC$_{50}$ of GEM alone (red) and GEM in combination with three sub-EC$_{50}$ doses of ATR inhibitor (shades of gray). (C) Cell viability of patient-derived organoids (PDO) UME008 and UME012 after treatment with GEM alone (red), GEM in combination with three sub-EC$_{50}$ doses of Elimusertib (shades of gray), or Elimusertib alone (blue dotted) relative to vehicle. (D) EC$_{50}$ of GEM alone (red) and GEM in combination with three sub-EC$_{50}$ doses of Elimusertib (shades of gray). Data information: Data are presented as mean values, with error bars representing the ± s.d. of triplicates (A and B) or eight replicates (C and D; except for UME012, where $n = 4$). In (B and D), asterisks show the significance level from a Student's t-test against GEM monotherapy (p-values were adjusted using the Benjamini–Hochberg procedure; *$p_{adj} < 0.05$; **$p_{adj} < 0.01$; ns: non-significant). Adjusted p-values are shown only for the most significant combinations; for all others, see Dataset EV5 (B) and Dataset EV8 (D). Source data are available online for this figure.

where the EC$_{50}$ for Elimusertib shifted from 58 nM to < 2 nM when combined with GEM (Fig. 1D).

We next verified the observed synergy of Elimusertib and GEM in AsPC-1 cells by cell growth assays (Fig. 2A) and including three additional clinical ATRi that were not part of the initial screen: Berzosertib, Ceralasertib, and Gartisertib. As single agents, Elimusertib and Gartisertib were already rather potent (EC$_{50}$ of 113 nM and 128 nM, respectively), while Berzosertib and Ceralasertib were much weaker (EC$_{50}$ of 766 nM and 1720 nM, respectively; Dataset EV5). Then, we turned the drug combination design around and titrated GEM (1 nM to 3,000 nM) in the presence of three sub-EC$_{50}$ doses of the ATR inhibitors and measured the shift in potency compared to GEM alone (EC$_{50}$ of 45 nM, mean of four experiments; Dataset EV5). Of the four ATR inhibitors, Elimusertib and Gartisertib sensitized

AsPC-1 cells most strongly to GEM (EC$_{50}$ of 6 nM and 7 nM, respectively), followed by Berzosertib and Ceralasertib (EC$_{50}$ of 18 nM and 12 nM, respectively; Fig. 2B; Dataset EV5). The same experimental setup was repeated in cell lines PANC-1 and Suit2-07, where all four drug combinations demonstrated superior efficacy compared to GEM monotherapy (Fig. EV2A,B; Dataset EV6). Importantly, cells were sensitized to GEM at extremely low doses of ATRi (sub-EC$_{10}$). These results demonstrate that the observed synergy of Elimusertib and GEM in the initial PDAC cell line screen can also be generalized to other ATR inhibitors and, therefore, underscores the involvement of the DNA repair machinery of the cell via ATR kinase. To determine whether drug synergy is specific to GEM only, Elimusertib was also evaluated in combination with other commonly used chemotherapeutic agents in AsPC-1 cells. Elimusertib sensitized AsPC-1 cells to

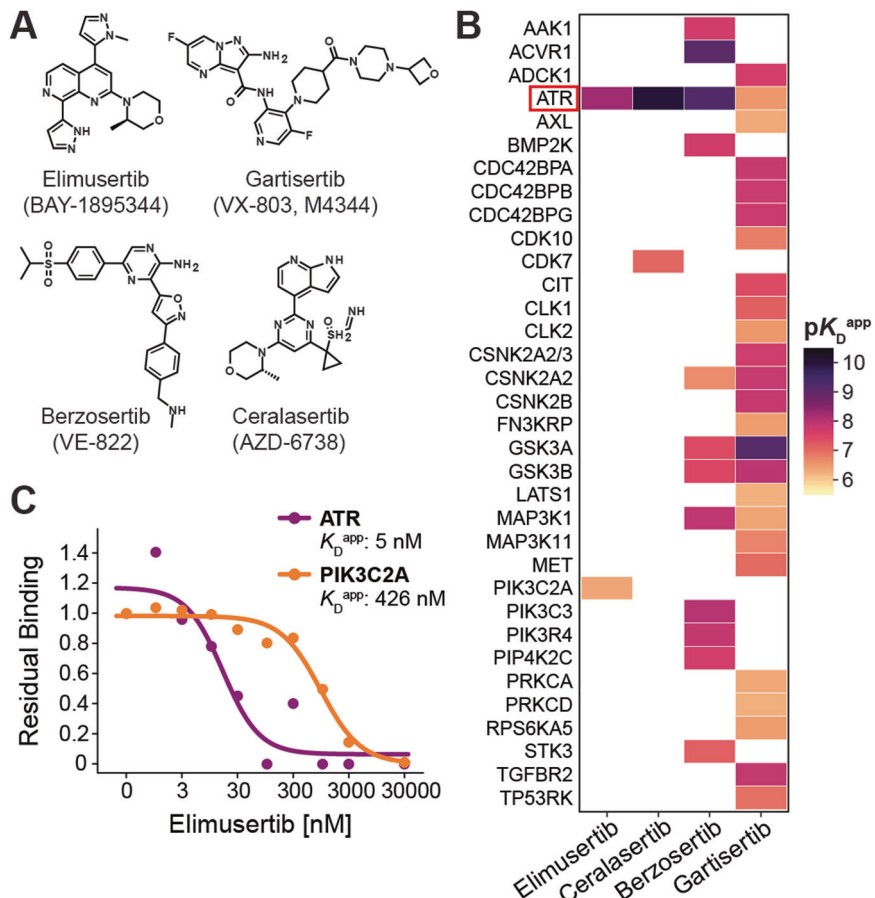

**Figure 3.  Chemoproteomic selectivity profiling of ATR inhibitors confirms ATR as the only common target kinase.**

(A) Molecular structures of the four clinical ATR inhibitors Elimusertib, Gartisertib, Berzosertib, and Ceralasertib. (B) Heatmap displaying the binding affinities ($pK_D^{app}$) for all target kinases identified with the Kinobeads technology. The designated target kinase ATR is highlighted with a red rectangle. (C) Residual binding of the three kinases ATR, PI3KCB, and PIK3C2A on Kinobeads upon increasing doses of Elimusertib. Binding affinities for each target kinase are given in the legend. Source data are available online for this figure.

the three DNA-damaging agents Oxaliplatin, Cisplatin, and 5-fluorouracil, but not to Paclitaxel (Fig. EV2C,D; Dataset EV7). This suggests that drug synergy with ATRi is not limited to GEM but also applies to other DNA-damaging agents, further highlighting the critical role of disrupted DNA repair in the synergy mechanism.

To evaluate the synergistic effects in more advanced disease models, nine PDAC patient-derived organoids (PDOs) were tested for their response to GEM combined with ATRi Elimusertib. Sensitivity to the individual drugs as monotherapies varied among the PDOs, with $EC_{50}$ values ranging from 3 nM to 285 nM for GEM and from 43 nM to 290 nM for Elimusertib (Dataset EV8). Upon the addition of sub-$EC_{50}$ doses of ATRi (30 nM or lower), five out of nine PDOs were significantly sensitized to GEM in at least one treatment condition (> 2-fold reduction in $EC_{50}$ and adjusted $p$-value < 0.05 compared to GEM alone; Figs. 2C,D and EV2E; Dataset EV8). The remaining four PDOs remained mostly unaffected. These results suggest that this regimen may be efficacious only in specific biological contexts, underscoring the importance of identifying predictive response markers for GEM and ATRi.

## Chemoproteomic target deconvolution reveals the selectivity of clinical ATR inhibitors

The strong chemosensitizing effect of the four tested ATR inhibitors points to the inhibition of ATR kinase activity as the molecular mechanism underlying the observed synergy. Because kinase inhibitors often have more than one target, we subjected Elimusertib and the three investigational ATRi Gartisertib, Berzosertib, and Ceralasertib to chemoproteomic target deconvolution in AsPC-1 lysates using the Kinobeads approach (Reinecke et al, 2019) (Fig. 3A) to exclude the effect of an unknown common off-target among the four inhibitors. As expected, all four drugs potently bound ATR kinase and, importantly, ATR was the only common target protein between the four (Figs. 3B and EV3A,B; Dataset EV9). The Kinobeads data did not reveal any interaction with the structurally and functionally related kinases ATM, DNAPK, or mTOR (Blackford and Jackson, 2017) (Dataset EV9). However, all four compounds showed off-target binding but to greatly different extents. In contrast to previous findings (Zenke et al, 2019), we observed that Gartisertib is a multi-kinase inhibitor

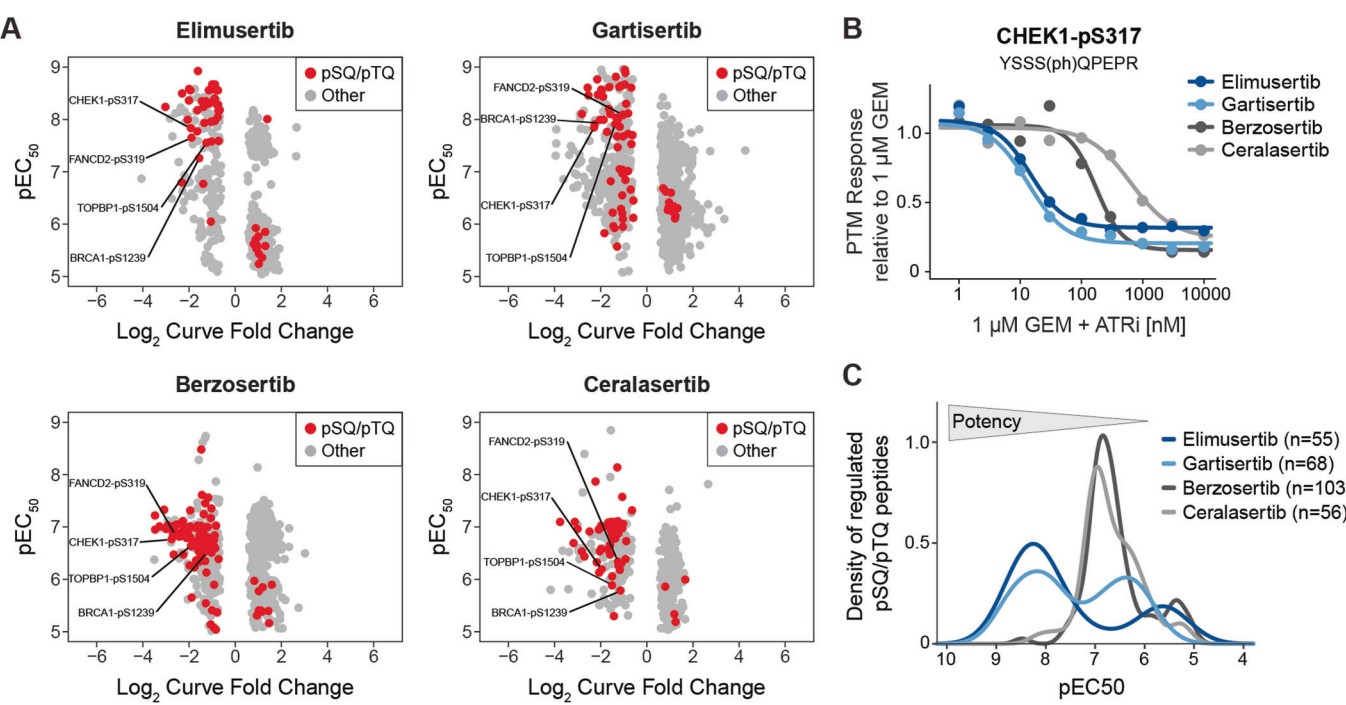

**Figure 4. DecryptM reveals cellular engagement of ATR kinase.**

(A) Potencies (pEC$_{50}$) and curve fold changes (log$_2$) from phosphoproteomic decryptM experiments with Elimusertib and Gartisertib (top) and Berzosertib and Ceralasertib (bottom) in DNA-damaged AsPC-1 cells acquired by LC-MS/MS. Each dot represents one dose–response curve of a regulated p-peptide (red: pSQ/pTQ, gray: non-pSQ/pTQ). Known direct substrates of ATR are annotated by text. (B) Dose-dependent inhibition of CHEK1-pS317 by four ATR inhibitors in DNA-damaged AsPC-1 cells. PTM response was normalized to 1 µM GEM. (C) Density plot summarizing the drug potencies (pEC$_{50}$) by which the four ATRi regulate the phosphorylation of pSQ/pTQ-peptides in DNA-damaged AsPC-1 cells. The number of pSQ/pTQ-peptides regulated per drug is given in the legend. Source data are available online for this figure.

without selectivity for ATR. Indeed, the drug showed stronger interactions with 17 kinases beyond its designated target ATR ($K_D^{app}$ = 318 nM), including GSKA and GSKB ($K_D^{app}$ < 10 nM), and several subunits of CSNK2 ($K_D^{app}$ = 10–20 nM). Gartisertib should, therefore, not be applied as a tool compound for investigating ATR-related cellular effects. Berzosertib also displayed several off-targets in the assay, but ATR was the most potent target by > 10-fold, with the exception of ACVR2 (ATR and ACVR2, $K_D^{app}$ < 1 nM vs. all others $K_D^{app}$ > 10 nM). For Elimusertib (Fig. 3C) and Ceralasertib, only one off-target each was observed, and the affinity to ATR over these off-targets was ~80-fold (for Elimusertib: ATR, $K_D^{app}$ = 5 nM vs. PIK3C2A, $K_D^{app}$ = 426 nM; for Ceralasertib: ATR, $K_D^{app}$ < 1 nM vs. CDK7, $K_D^{app}$ = 84 nM). Together, this verifies that the synergistic phenotypic effect of the four drugs can be explained by the inhibition of ATR.

## Clinical ATR inhibitors engage phosphorylation-regulated signaling pathways

To investigate if and how the above ATR inhibitors engage their targets in cells, we followed the decryptM approach (Zecha et al, 2023) and performed mass spectrometry-based phosphoproteomics on GEM-induced DNA-damaged and subsequently ATRi-treated cells (Fig. EV4A). Briefly, AsPC-1 cells were pre-incubated with a high dose of GEM (1 µM) for 3 h to induce DNA damage followed by nine doses of ATRi (1 nM to 10 µM) for 1 h (see Methods for

details). Across all experiments, 25,537 non-redundant phosphorylated peptides were quantified, of which 20,784 had phosphorylation sites (p-sites) that could be localized with a probability of > 0.75 (15,014 to 17,314 p-sites per experiment; Dataset EV10). Only the latter were used for all subsequent analyses. The underlying ~64,000 p-site dose–response profiles were statistically evaluated and categorized into significant up- and down-regulated curves by CurveCurator (Bayer et al, 2023) (see Methods for details). As one might expect from the Kinobeads data above, the decryptM analysis uncovered dose-dependent regulation of more p-sites for Berzosertib ($n = 840$) and Gartisertib ($n = 1015$) compared to the more selective compounds Ceralasertib ($n = 421$) and Elimusertib ($n = 525$), likely as the result of inhibiting off-targets in cells. Among the significantly inhibited p-sites, there was a striking over-representation of p-sites with a pSQ/pTQ motif (Figs. 4A and EV4B), which is known to be the kinase substrate motif of ATR. Other atypical kinases, namely ATM and DNAPK, share the same motif (Kim et al, 1999), but they can be excluded as the upstream kinase because they are not targeted by any of the drugs. Among these sites, the well-known ATR substrate and key effector site CHEK1-pS317 (Zhao and Piwnica-Worms, 2001) was inhibited in a dose-dependent fashion by all four ATRi, which clearly demonstrates reduced kinase activity in cells (Fig. 4B). Inhibition of ATR signaling further manifested in suppressed phosphorylation of the ATR-activator TOPBP1 (Kumagai et al, 2006) (pS1504), the homologous recombination mediator BRCA1

(Tibbetts et al, 2000) (pS1239) as well as FANCD2 (pS319), which is involved in the repair of cross-linked DNA upon phosphorylation by ATR (Kupculak et al, 2023). The dose-dependent decryptM approach revealed that the four drugs inhibited cellular DNA damage signaling (via pSQ/pTQ-sites) at different potencies. When summarizing $EC_{50}$ values across all drug-regulated pSQ/pTQ-sites, Elimusertib and Gartisertib were 10- to 20-fold more potent in cells (median $EC_{50}$ of 10 nM and 21 nM, respectively) than Ceralasertib and Berzosertib (median $EC_{50}$ of 154 nM and 188 nM, respectively; Fig. 4C). This mirrors the phenotypic cell viability data presented above for the same drugs. We note that the Kinobeads binding affinity data collected in cell lysates would have suggested the opposite trend. This discrepancy may be explained by differences in cellular uptake or export of the different drugs.

Because ATR was confirmed as the only common target of the four inhibitors, we declared p-sites regulated by at least three of the four ATRi as bona fide (direct or downstream) ATR-dependent phosphorylation events. This resulted in 298 regulated p-peptides, 42 of which were pSQ/pTQ-sites and thus likely direct substrates of ATR (Fig. EV4C, Dataset EV10). Among the regulated non-pSQ/pTQ-sites were > 30 known direct substrates of cyclin-dependent kinases (CDK1 and CDK2; based on kinase-substrate annotations from PhosphoSitePlus (Hornbeck et al, 2015); Dataset EV10), linking ATR pathway engagement to subsequent impaired cell cycle control. Regulations in this category encompassed p-sites on the cell cycle regulator CHEK1 (pS268), the spindle assembly proteins TPX2 (pT72) and PRC1 (pT481), as well as the proliferation marker MKI67/Ki67 (pT761; Dataset EV10).

## ATRi potently block GEM-induced DNA damage response, explaining drug synergy

To pinpoint the cellular mechanism underlying the observed synergy of GEM and ATR inhibition, we took a step back and analyzed the phosphoproteomic data from the angle of which p-sites were regulated by GEM treatment alone (representing induction of DNA damage), and reversed by ATR inhibition in the combination treatment. Treatment of cells with 1 μM GEM alone vs. untreated cells revealed 414 statistically significant GEM-regulated p-peptides (216 up, 198 down), including 135 pSQ/pTQ motif sites (adjusted $p$-value < 0.01, $\log_2$ fold change > 1, quantified in at least three out of four replicates; Fig. 5A; Dataset EV10). As expected, pSQ/pTQ-sites were mostly increased upon administration of the DNA-damaging agent and accounted for 62% of all upregulation events (134 out of 216 p-peptides; Fig. EV5A). ATR inhibition countered 164 of the 416 GEM-induced phosphorylation changes including 36 pSQ/pTQ-sites (Figs. 5A and EV5B). More specifically, ATRi reverted GEM-induced phosphorylation (62 p-peptides including the 36 pSQ/pTQ; Fig. EV5C), and restored GEM-inhibited phosphorylation (102 p-peptides, only non-pSQ/pTQ; Dataset EV10). Phosphorylation of several CHEK1 p-sites was suppressed upon combination therapy, including the aforementioned CHEK1-pS317 ATR substrate and CHEK1-pS286 (a CDK substrate). The most strongly affected p-site was CHEK1-pS468 (pSQ/pTQ). Phosphorylation of this site was increased > 18-fold upon GEM treatment ($\log_2$ fold change of 4.2, adjusted $p$-value < 0.002) and reversed up to 14-fold by three of the four ATRi (not quantified for Gartisertib; $\log_2$ fold change upon ATRi between −3.8 and −2.9; Fig. 5B,C). Interestingly, despite CHEK1

being a well-known effector protein of ATR, this particular p-site has not yet been studied in depth in the context of ATR signaling (Zhao and Piwnica-Worms, 2001). However, our data suggests that CHEK1-pS468 is a clear marker of ATR activity and drug synergy and, thus, may be used as a response biomarker in future investigations in animal studies or clinical trials.

To place our findings in the context of cellular signaling, we performed pathway enrichment and visualization using PTMNavigator (Müller et al, 2025). As expected, ATR signaling (WikiPathways entry WP4016) was among the most highly enriched pathways (enrichment score 14.51; Fig. 5D). The proteins annotated in this signaling pathway contained nine of the 36 ATRi-blocked pSQ/pTQ sites, including the ATR substrates CHEK1 (pS317, pS468), TOPBP1 (pS1504), and FANCD2 (pS319). Interestingly, a number of proteins and p-sites within this pathway were affected by GEM in both an ATR-dependent and ATR-independent fashion. For instance, although GEM-induced DNA damage led to increased phosphorylation of the homologous recombination (HR) protein BRCA1 at seven p-sites, only two of these were reverted by ATRi (pSQ/pTQ pS1239 and non-pSQ/pTQ pS1642; Fig. 5D). Other proteins involved in HR-mediated DNA repair were also partially affected by the drug combination (depending on the site). These included the BRCA1 interactor UIMC1 (also known as RAP80) (Yan et al, 2007) and NBN (also known as NBS1), which is essential in the repair of DNA double-strand breaks via HR (Tauchi et al, 2002). Our analysis also revealed proteins which were throughout affected in phosphorylation exclusively by GEM, but not by ATRi on any site. These comprised H2AX (pS139), a well-established DNA damage marker (Rogakou et al, 1998), and additional pSQ/pTQ-sites on MDC1 (pS955, pS1086) and TP53BP1 (pS831). These proteins are commonly described as taking part in ATM-mediated DNA repair and cell cycle control (Kim et al, 2006; Stucki and Jackson, 2006). Hence, we suspect that these GEM-induced effects arise from ATRi-independent DNA damage signaling, presumably by the other master regulator kinase ATM. Because the four drugs investigated here do not inhibit ATM, these GEM-only p-sites do not relate to the observed drug synergy. This is also in agreement with our observation that ATM inhibition (via AZD-0156) did not synergize with GEM in the initial viability screen (median $\Sigma AUC_{\Delta}$ across all cell lines < 2%; Dataset EV2). Overall, the above analysis demonstrates the potential of the decryptM approach to discriminate ATR-dependent from ATR-independent signaling in response to GEM-induced DNA damage. It also highlights the importance of p-site level-resolved analysis to understand the mechanism of drug synergy.

The majority of pSQ/pTQ-sites we declared to explain drug synergy were not covered by the currently annotated ATR signaling network shown above (> 20 sites). These include proteins described to participate in other DNA damage repair processes, and our data suggests a connection to ATR kinase in these pathways as well (Dataset EV10). Among them were highly regulated pSQ/pTQ-sites on HMGA1 (pS9, pS44), PPM1G (pS201), and TSEN34 (pS136; all > 4-fold induced by GEM and inhibited by ATRi; Figs. 5C and EV5C). However, phosphorylation changes also affected proteins that currently lack a clear association to ATR signaling or DNA damage repair. Intriguingly, some of these proteins contain some of the most strongly GEM-induced pSQ/pTQ-sites in the dataset: the cytosine methylase NSUN5-pS432 (7-fold), the

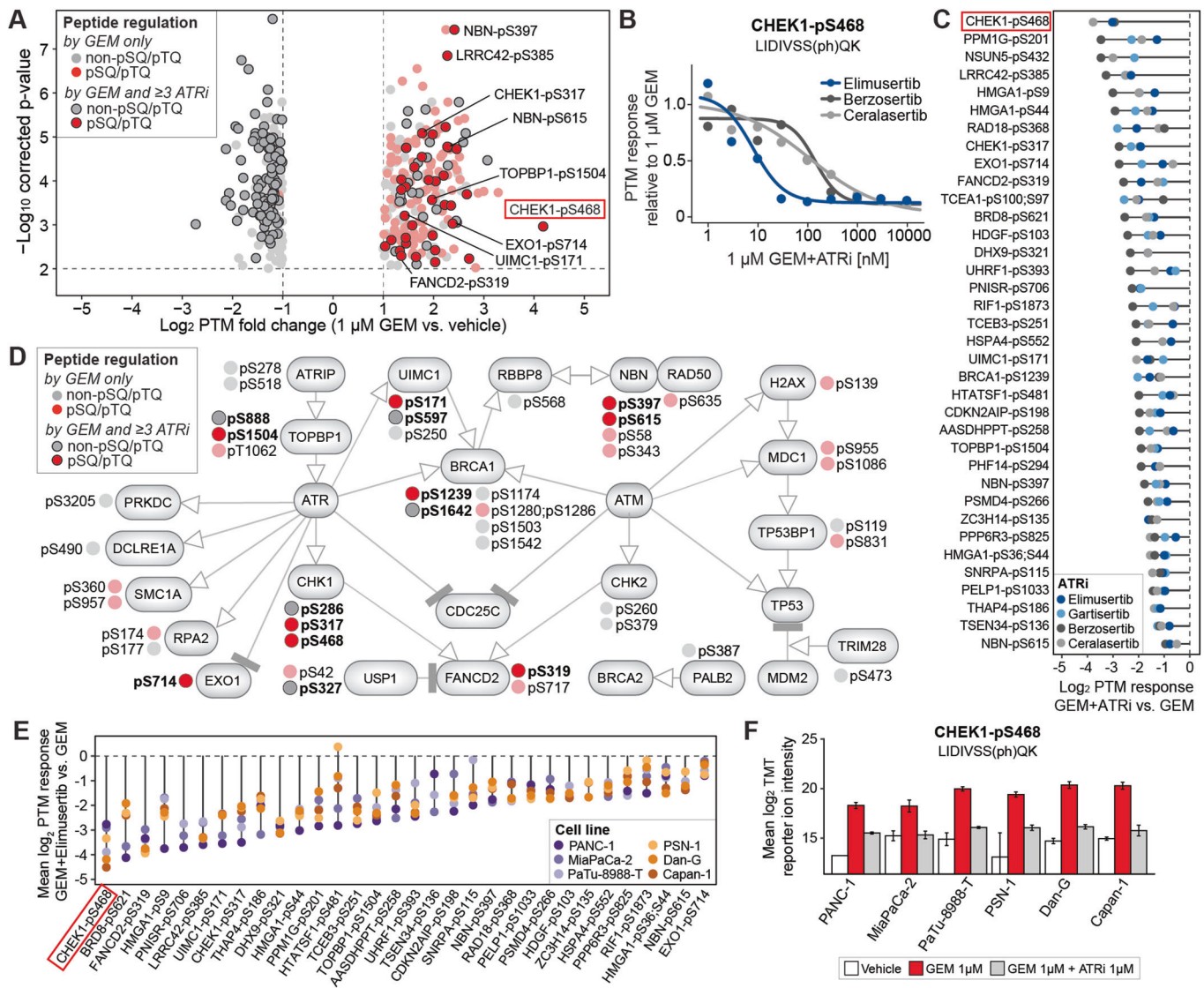

**Figure 5. Partial blockage of GEM-induced DNA damage signaling by ATRi explains drug synergy.**

(A) Volcano plot of 1 μM GEM vs. vehicle in AsPC-1 cells after 4 h (Student's t-test, $n = 4$ independent experiments). P-values were adjusted using the Benjamini–Hochberg procedure. Phosphorylated peptides (red: pSQ/pTQ; gray: non-pSQ/pTQ) were either regulated by GEM only (lighter shades), or by GEM and at least three ATRi (darker shades, encircled). Selected peptides are annotated by text. (B) Dose-dependent inhibition of CHEK1-pS468 by three ATRi in DNA-damaged AsPC-1 cells. PTM response was normalized to 1 μM GEM. (C) PTM response ($\log_2$) of 36 pSQ/pTQ sites upon GEM and ATRi in AsPC-1 cells relative to GEM alone. (D) Selected nodes of the WikiPathway WP4016, which was most enriched in PTMNavigator (Score = 14.51). Annotated p-sites (red: pSQ/pTQ; gray: non-pSQ/pTQ) were regulated either by GEM only (lighter shades) or by GEM and ATRi (darker shades, encircled). (E) Mean PTM response ($\log_2$) of 32 of the 36 pSQ/pTQ sites upon 1 μM GEM and 1 μM Elimusertib in six additional PDAC cell lines, relative to 1 μM GEM ($n = 4$). (F) Mean TMT reporter ion intensity ($\log_2$) of CHEK1-pS468 in six PDAC cell lines treated with vehicle, 1 μM GEM, or 1 μM GEM plus 1 μM Elimusertib. Error bars indicate the ± s.d. of quadruplicates ($n = 4$). Source data are available online for this figure.

leucine-rich protein LRRC42-pS385 (4.8-fold) and the splicing factor PNISR-pS706 (4.6-fold increase upon GEM and mitigated by at least three ATRi; Figs. 5C and EV5C). Out of these, the strongest combination effect was seen for LRRC42-pS385, which was fully blocked to baseline levels by all four ATRi (Fig. EV5D). Based on these observations, we conclude that (apart from CHEK1-pS468 mentioned above), these sites may play an important functional role in DNA damage and (potentially non-canonical) ATR signaling, and could serve as drug-response markers for the combination of GEM and ATRi.

To assess the robustness of the proposed biomarkers, we examined their regulation in six additional PDAC cell lines (PANC-1, MiaPaCa-2, PaTu-8988-T, PSN-1, Dan-G, and Capan-1) treated with 1 μM GEM, 1 μM GEM plus 1 μM Elimusertib, or vehicle. Of the 36 pSQ/pTQ-sites, 32 were quantified, with the majority showing > 2-fold reduction upon combination treatment compared to GEM alone (Fig. 5E; Dataset EV11). Notably, strong inhibition was observed for the known ATR substrates CHEK1-pS317 and FANCD2-pS319, as well as the novel substrate candidate LRCC42-pS385 ($\log_2$ fold changes < −2 in at least three cell lines).

Consistent with the AsPC-1 results, CHEK1-pS468 was most strongly inhibited, with $\log_2$ fold changes as low as −4.5 and −4.2 in Capan-1 and Dan-G cells (adjusted $p$-values $< 8e^{-3}$ and $< 4e^{-3}$, respectively; Fig. 5F; Dataset EV11). Importantly, known substrates of the related kinase ATM, such as ATM-pS1981, H2AX-pS139, BRCA1-pS1524, and TRIM28-pS824 (all pSQ/pTQ), remained unaffected by ATRi across all cell lines, further suggesting ATR-specific regulation (Appendix Fig. S1). These results highlight the consistent regulation of the proposed pSQ/pTQ-sites in response to GEM and ATRi, underscoring their potential as biomarkers for treatment response.

Additionally, we have mined phosphoproteomic PDAC patient data as part of the CPTAC project, focusing on the 36 pSQ/pTQ-sites linked to drug synergy. Eight of these sites were detected in patients, with all but one showing elevated phosphorylation levels in PDAC tumors compared to adjacent normal tissue (Appendix Fig. S2A). Moreover, protein expression analysis indicated that although ATR is not differentially expressed in PDAC, tumor cells exhibit significantly higher CHEK1 levels than normal tissue (Appendix Fig. S2B). While no formal proof, these data imply that there may be increased DNA damage response activity in tumor cells, which may be repressed by ATR inhibition.

## Discussion

Because of the very poor prognosis of most PDAC patients, there is a desperate medical need for more successful therapeutic approaches. The combination of standard chemotherapy such as GEM with molecularly targeted drugs may be a valuable strategy but needs a strong molecular rationale to be meaningfully tried in the clinic. Because only few drug combination screens have been performed in the context of PDAC in pre-clinical research settings (Falcomata et al, 2022; Jaaks et al, 2022; Nair et al, 2023; O'Neil et al, 2016; Zhang et al, 2023), the current work provides a substantial new molecular resource to provide such a rationale.

One important finding was that drug synergy is rare, which is in line with earlier published studies (Jaaks et al, 2022; Nair et al, 2023; O'Neil et al, 2016). Specifically, the lack of synergy with RTKi in vitro indicates that such combinations may only be clinically successful in special and rare cases, raising the need for biomarkers enabling patient stratification.

The second, and more encouraging, key finding was that the ATR inhibitor Elimusertib strongly sensitized almost all PDAC cells to GEM. With hindsight, this may not be unexpected as GEM induces DNA damage, and ATR plays a central role in several aspects of the cellular DNA damage response, notably in detecting and repairing DNA damage during cell replication to ensure genomic stability (Cimprich and Cortez, 2008). In very recently published work, Zhang et al investigated combined effects of DNA damage response kinase inhibitors, including three ATRi (not Elimusertib), in 62 cancer cell lines covering twelve tumor types (Zhang et al, 2023). They also found ATRi and GEM to synergize across several cancer cell lines (including four PDAC), and reported ATRi to show stronger overall synergy compared to other DNA damage inhibitors. This is in line with our findings for Elimusertib, and further indicates that the combination of ATRi and GEM may be a promising therapy for other cancer types as well. Although we are not aware of any other large-scale study that systematically

assessed drug synergy between ATRi and GEM, others have observed synergy between GEM and inhibitors of CHEK1 and WEE1, two cell cycle regulators that act downstream of ATR in response to DNA damage (O'Connell et al, 1997). For instance, Jaaks et al tested the combination of 20 targeted drugs and GEM across 30 PDAC cell lines and observed synergy with the CHEK1 inhibitor AZD7762 in nearly half of the cell lines and, to a lesser extent, also with the WEE1 inhibitor Adavosertib (Jaaks et al, 2022). Two further studies reported broad activity of GEM in combination with CHEK1 inhibition in NSCLC cells (Nair et al, 2023), or CHEK1i and WEE1i across several cancer entities other than PDAC (O'Neil et al, 2016). In our screen, WEE1i Adavosertib, as well as Brigatinib and Lestaurtinib, two broad-spectrum inhibitors with off-target activity for CHEK1, also displayed synergy with GEM in nearly half of our PDAC cell line panel. In contrast, the ATR inhibitor Elimusertib synergized with GEM in almost all cell lines. One could, therefore, speculate that the combination of GEM with ATR inhibition may be therapeutically more efficacious in a wider range of heterogeneous PDAC tumors (or other entities) than targeting kinases downstream of ATR. Some evidence for such an interpretation also comes from our single drug screening data, in which the efficacy of Elimusertib was superior to that of Adavosertib, Brigatinib, or Lestaurtinib.

The dose-dependent phosphoproteomic profiling of Elimusertib following induction of DNA damage by GEM, together with the fact that none of the tested ATR inhibitors showed inhibition of ATM, clearly indicated that the observed drug synergy is rooted in the inability of cells to repair DNA via ATR-dependent mechanisms. For translational purposes, and to be able to link a therapeutic response to the molecular mechanism of a drug, an adequate pharmacodynamic biomarker is required. In our study, we discovered 36 pSQ/pTQ motif p-sites that could be used to monitor drug response. More specifically, these p-sites were strongly induced by GEM and potently blocked by ATRi. The dataset also confirmed earlier reports that known ATR substrates, including CHEK1, are involved in drug synergy (Liu et al, 2017; Wallez et al, 2018). The very robust signal observed for CHEK1-pS468 (pSQ/pTQ; 18-fold induced by GEM and fully countered by ATRi) in the current study makes this p-site particularly promising as a mechanistic marker for the response to the GEM-Elimusertib combination. Interestingly, while CHEK1 is a well-studied protein in DNA damage signaling, very little is known about CHEK1-pS468. In fact, the authors are only aware of a single study that investigated ATR-dependent phosphorylation of this site some 20 years ago (Zhao and Piwnica-Worms, 2001). Another pharmacodynamic biomarker candidate is the methyltransferase NSUN5-pS432 (pSQ/pTQ, 7-fold increase by GEM and neutralized by three ATRi). Although the evidence from the literature is sparse, this site was described to be regulated by ATM kinase in response to DNA double-strand break before (Larsen and Stucki, 2016). Here, we find that this site is likely a direct substrate of ATR and a promising marker of drug synergy. Finally, another promising marker candidate is LRRC42-pS385 (pSQ/pTQ; ~5-fold increase by GEM and fully blocked by all four ATRi). LRRC42 is a largely uncharacterized protein. The few cancer-related reports for the protein describe a potential role in lung carcinogenesis and its overexpression in breast tumors, but no connection to DNA damage response signaling has been made yet (Fujitomo et al, 2014; Moody et al, 2020). To the best of our knowledge, phosphorylation

of this site has also not been reported yet, let alone its induction by DNA damage or in response to drugs.

MS-based phosphoproteomics has been used before to study cellular signaling in response to DNA damage induced by radiation or chemotherapy (Bensimon et al, 2010; Matsuoka et al, 2007; Mu et al, 2007). The current work goes substantially beyond the state of the art in several aspects. First, the phenotypic drug screen across many cell lines revealed that ATR inhibition might be a general mechanism by which PDAC cells could be sensitized to GEM. In turn, this may provide a more general way to break GEM resistance in PDAC or other cancers. Second, the study also illustrates that the application of drug dose-dependent chemical proteomic approaches for target identification and selectivity profiling (Kinobeads), as well as pathway engagement measurements (decryptM) greatly facilitates the elucidation of the molecular basis for the observed synergistic drug effects in cells. Third, and aided by the obtained broad (phospho-)proteomic coverage, novel ATR substrates could also be identified, including drug response markers that may turn out to be useful for translational studies in the future. In particular, these need to address if the synergy observed for the combination of GEM and ATR inhibitors in 2D cell lines and PDOs also translates to more sophisticated PDAC models, such as patient-derived xenografts (PDX). These experiments will also be valuable to test the suitability of, for instance, the aforementioned CHEK1-pS468 phosphorylation as a patient stratification or therapeutic response marker.

Recently, Jadav et al investigated the MoA of Berzosertib and Gartisertib in US-O2 osteosarcoma cells treated with hydroxyurea (Jadav et al, 2024). Despite many differences in, for instance, the drug used for induction of DNA damage, the cellular model, and the phosphoproteomic approach applied, individual drug-regulated p-sites are shared between both studies. This partial overlap in the quantified ATR signaling strengthens our confidence in the robustness of our phosphoproteome data. In addition, our work extends the available phosphoproteomic datasets on ATR and DNA damage signaling in cells, and unveils a mechanistic rationale for clinically relevant drug combinations.

Beyond the focus of this report on discovering the basis for the synergy of GEM and Elimusertib, the provided data can be used for several additional purposes. For instance, the dose-dependent single drug screen data was not systematically explored in this report. The same is true for most of the synergistic combinations that affected only a few cell lines. In this context, baseline (phospho-)proteomes and kinase activity landscapes of the cell line panel could further help prioritize promising drug combinations, as previously demonstrated by Vallés-Martí et al (2023). In addition, the many p-sites that were induced by GEM and reverted by ATR inhibition are not yet functionally understood, particularly regarding their role in DNA damage response. For instance, it is possible that some of the ATR inhibitor-regulated pSQ/pTQ p-sites actually arise from ATM activation as a result of ATR inhibition (Wallez et al, 2018). Eventually, the classification of ATR-independent and ATR-dependent DNA damage response on a single site-resolved level may be explored to better understand the previously reported crosstalk of these kinases (Fedak et al, 2021). In future work, phosphoproteomic studies using ATM inhibitors on GEM-treated cells could complement the current work as these may identify further molecular mechanisms that are based on DNA damage, helping to pinpoint overlapping and distinct substrates that contribute to GEM-ATRi synergy. Moreover, while our study identified disrupted DNA damage response as the primary mechanism of drug synergy, there is growing evidence of non-canonical activities of ATR that may contribute to these combination effects (da Costa et al, 2023). Notably, prolonged exposure of PDAC cells to GEM and ATRi resulted in differential expression of enzymes involved in nucleotide metabolism, specifically RRM2 (but not RRM1) and TK1 (Appendix Fig. S3; Dataset EV12). Given that both ATR and GEM are known to impact cellular nucleotide pools, this dysregulation likely represents an additional mechanism driving GEM-ATRi synergy (da Costa et al, 2023; Huang et al, 1991). These findings prompt further investigation into such non-canonical ATR pathways, which may explain the superior synergy between GEM and ATRi compared to other DNA damage response inhibitors.

Notwithstanding these limitations of the current work, our findings provide a mechanistically rational explanation for the combination of GEM and ATR inhibitors in ongoing clinical trials for PDAC (Middleton et al, 2021; NCT03669601; NCT04616534). In addition, the results also show that Elimusertib produces superior cellular efficacy as a result of the most potent and selective target and pathway engagement among the four investigational ATR inhibitors tested. Hence, one might suggest prioritizing Elimusertib over other drugs for clinical trials investigating GEM-ATRi combinations.

# Methods

**Reagents and tools table**

| Reagent/Resource | Reference or Source | Identifier or Catalog Number |
|---|---|---|
| **Experimental models** | | |
| AsPC-1 (H. sapiens) | ATCC | CRL-1682 |
| Additional pancreatic cancer cell lines | See Appendix Table S1 | See Appendix Table S1 |
| Pancreatic patient-derived organoids | See Appendix Table S2 | See Appendix Table S2 |
| **Chemicals, Enzymes and other reagents** | | |
| Elimusertib | Selleckchem | S8666 |
| Gartisertib | Selleckchem | S9639 |
| Ceralasertib | Selleckchem | S7693 |
| Berzosertib | Selleckchem | S7102 |
| Gemcitabine | Selleckchem | S1714 |
| Oxaliplatin | Selleckchem | S1224 |
| Cisplatin | Selleckchem | S1166 |
| 5-Fluorouracil | Selleckchem | S1209 |
| Additional compounds used in viability screen | See Dataset EV1 | See Dataset EV1 |
| TMT10plex Isobaric Label Reagent Set | Thermo Fisher Scientific | 90406 |
| TMT11-131C Labeling Reagent | Thermo Fisher Scientific | A34807 |
| TMTpro 16plex Label Reagent Set | Thermo Fisher Scientific | A44520 |
| DMEM | PAN Biotech | P04-04510 |

| Reagent/Resource | Reference or Source | Identifier or Catalog Number |
|---|---|---|
| RPMI | PAN Biotech | P04-18500 |
| DMEM/Hams-F12 (1:1) | PAN Biotech | P04-41154 |
| Fetal Bovine Serum | PAN Biotech | P04-18500 |
| HEPES | PAN Biotech | P05-01100 |
| Non Essential Amino Acid Solution | PAN Biotech | P08-32100 |
| Bovine Pituitary Extract | Thermo Fisher Scientific | 13028014 |
| Recombinant human EGF protein | R&D Systems | 236-EG |
| Keratinocyte-free medium | Thermo Fisher Scientific | 17005042 |
| Penicillin Streptomycin GlutaMAX | Thermo Fisher Scientific | A5873601 |
| Reagents used for patient-derived organoids | See Appendix Table S3 | See Appendix Table S3 |
| ATPlite 1step Luminescence Assay System | revvity, previously Perkin Elmer | 6016731 |
| CellTiter-Glo Luminescent Cell Viability Assay | Promega | G7572 |
| **Software** | | |
| MaxQuant (v1.6.12.0, v2.1.3.0) | Cox and Mann (2008) | |
| Perseus (v1.6.2.3) | Tyanova et al (2016) | |
| SIMSI-Transfer (v0.5.0) | Hamood et al (2022), https://github.com/kusterlab/SIMSI-Transfer | |
| CurveCurator (v0.4.1) | Bayer et al (2023), https://github.com/kusterlab/curve_curator | |
| PTMnavigator | Müller et al (2025), https://www.proteomicsdb.org/analytics/ptmNavigator | |
| **Other** | | |
| Dionex UltiMate 3000 RSLCnano System | Thermo Fisher Scientific | |
| Fusion Lumos Tribrid mass spectrometer | Thermo Fisher Scientific | |
| Janus Gripper | revvity, previously Perkin Elmer | |
| Multidrop Combi Reagent Dispenser | Thermo Scientific | |
| Incucyte S3 | Sartorius | |
| Envision Xcite 2104 plate reader | revvity, previously Perkin Elmer | |
| Spark Multimode microplate reader | TECAN | |
| VantaSTAR | BMG Labtech | |

## Cell lines and patient-derived organoids

Cell lines were purchased from ATCC (MiaPaCa-2, AsPC-1, PSN-1, BxPC-3, PANC-1), CLS (FAMPAC, Capan-1), DSMZ (PaTu-8988-T, Dan-G) or creative bioarray (HuP-T4). Cell lines Suit2-07, Colo-357, and HPDE were kindly provided by Prof Kirsten Lauber (LMU, Germany). The identity of all cell lines was authenticated using STR fingerprinting (Suit2-07) or SNP profiling (all others) as provided by Multiplexion GmbH, Germany. Cells were cultivated at 37 °C and 5% $CO_2$ and regularly checked to be mycoplasma-free. Detailed information on cell culture media is provided in Appendix Table S1. All media and supplements were purchased from PAN biotech, except for human epidermal growth factor (R&D Systems), bovine pituitary extract (Gibco), and keratinocyte-free medium (Gibco). Only for the drug combination screen, cell media were supplemented with Pen-Strep (100 U/ml penicillin and 100 µg/ml streptomycin, Gibco).

Primary patient-derived PDAC organoid (PDO) experiments were performed in the laboratories of Prof. Maximilian Reichert (University Hospital of the TU Munich, Germany, PDO set 1: B250, B535, B678) and Prof. Jens T. Siveke (University Hospital Essen, Germany; PDO set 2: UME008, UME012, UME051, UME053, UME060, UME061). PDOs were generated from surgical resection, EUS-guided fine needle biopsy (EUS-FNB), or ascites (Appendix Table S2). Surgical specimens were first minced with a sterile scalpel and subsequently washed with Anti-Anti solution and washing medium (PDO set 1), or vice versa (PDO set 2; Appendix Table S3). Biopsy samples were directly washed with an Anti-Anti solution and washing medium. Samples were digested using 1 ml of digestion medium at 37 °C while shaking, and the enzymatic reaction was stopped by adding 15 ml of washing medium (Appendix Table S3). If samples of PDO set 1 were blood-stained, tissue pellets were first incubated with ACK lysing buffer/red blood cells lysis buffer (Thermo Fisher Scientific) for 5 min and then digested using TrypLE (Thermo Fisher Scientific). The reversed procedure was applied for blood-stained samples of PDO set 2. For PDO generation from ascites, the material was first spun down at $500 \times g$ for 5 min. Then, washing medium was added and the suspension was centrifuged at $300 \times g$ for 5 min. If the pellet was blood-stained, ACK lysis buffer was added and incubated at room temperature. Samples were washed with washing medium, and pellets from PDO set 1 were resuspended in 200 µl of 10% BME (Cultrex RGF Basement Membrane Extract, Type 2, Pathclear) plus PDAC PDO medium (Appendix Table S3) + 1:1000 Y-27632 (Biomol/Cayman) in a 48-well plate. Pellets from PDO set 2 were resuspended in BME (Matrigel RGF Basement Membrane Matrix, Corning), seeded as 10x domes of 30 µl per well in a 6-well plate, and then covered by PDAC PDO medium + 1:1000 Y-27632 (PeproTech).

Patient material used for this study was approved by the local ethics committees (Munich: project 207/15, Essen: 22-10622-BO), and written informed consent was obtained from the patient prior to the investigation. No blinding was performed.

## Compounds

Compounds used for cell viability screening were purchased from different vendors, including Selleckchem (Absource Diagnostics GmbH, Germany) and MedChemExpress (Hölzel Diagnostika Handels GmbH, Germany; Dataset EV1), and their identity was confirmed by their mass using LC-MS/MS at high resolution. Drugs used for additional experiments are listed in the Reagents and Tools

Table. Chemical structures of drugs shown in this work were created using ChemDraw (v23.0.1).

## Drug combination screen

Library drugs were serially diluted from 10 mM to 0.0002 mM in DMSO (11 doses, 1:3 dilution steps) using a Janus Gripper with 384-channel Modular Dispense Technology dispensing head (revvity, previously Perkin Elmer, MA, USA). Afterward, small amounts were transferred to 384-well intermediate plates (Greiner Bio-One GmbH, Germany). GEM stock solutions (1 mM, 3 mM, 10 mM, and 30 mM) in DMSO were prepared manually. For drug combination screening, 1000 to 5000 cells were seeded into 384-well plates (CulturPlate™, Perkin Elmer, MA, USA) using a Multidrop™ Combi Reagent Dispenser (Thermo Scientific, MA, USA) in 50 µl medium per well 24 h prior to drug treatment (seeding densities see Appendix Table S1). Compound intermediate plates were filled up with water (1:45.5 dilution) with a Multidrop™ Combi Reagent Dispenser, and cells were treated with 2.5 µl of library drug serial dilutions using a Janus Gripper. Then, either 2.5 µl vehicle (for single drug treatments) or 2.5 µl of fixed-dose GEM (for drug combination treatments) were added using a Multidrop™ Combi Reagent Dispenser. The final DMSO concentration in each well was 0.2%. The applied doses of GEM were selected based on the cell lines sensitivity towards GEM as single drug (1 nM and 3 nM GEM for cell lines FAMPAC, Dan-G, Capan-1, Colo-357, PSN-1, HPDE, HuP-T4, BxPC-3, Suit2-07; 10 nM and 30 nM GEM for cell lines PaTu-8988-T, AsPC-1, PANC-1, MiaPaCa-2; Dataset EV3). After 72 h of incubation at 37 °C and 5% $CO_2$, viability was measured using the ATPlite 1step Luminescence Assay System according to the manufacturer's protocol (Perkin Elmer, MA, USA) and the luminescence signals per well were detected with Envision Xcite 2104 plate reader with ultrasensitive luminescence detector (revvity, previously Perkin Elmer, MA, USA). Eight wells per plate without library drug served as negative controls, and sixteen wells per plate containing 5 µM Staurosporine served as positive controls. Control plates not containing any drug were used to correct for incubation effects, and data quality was assessed by calculation of Z-prime of positive and negative controls according to

$$Z\text{-prime} = 1 - \frac{3SD_{pos} + 3SD_{neg}}{|mean_{pos} + mean_{neg}|} \quad (1)$$

where SD is the standard deviation. All compounds were tested for media solubility with their final dilutions using a NEPHELOstar microplate reader (BMG LABTECH), and a counterscreen was performed to exclude interfering reaction of compounds with ATP instead of cells with ATPlite 1step Luminescence Assay System to detect false active compounds. All data was normalized to a vehicle (mean negative controls), and dose–response curves were fitted to a four-parameter log-logistic model, and area-under-curve (AUC, ranging between 0–100%) was calculated as described previously (Lee et al, 2024). Single drugs were considered efficacious if AUC < 80%, $pEC_{50} > 6$, and $R^2 > 0.8$.

## Calculation of drug synergy

Synergistic effects in the drug screen were estimated by applying the concept of Bliss Independence (Bliss, 1939). Therefore, a hypothetical dose–response curve was calculated by multiplying the response of the single drugs at each dose used in the combination, describing the expected response when no combination effect occurs, using the following formula:

$$E_{A+B,hypothetical} = E_{A,experimental} \cdot E_{A,experimental} \quad (2)$$

where $E_{A,experimental}$ and $E_{B,experimental}$ are the cell viability upon single-agent treatment with drugs A and B, respectively, and $E_{A+B,hypothetical}$ is the expected cell viability upon combination. Dose–response curves from the experimental and hypothetical data were compared, and shifts in AUC were used to describe combination effects ($AUC_\Delta = AUC_{hypothetical} - AUC_{experimental}$). Drug combinations were defined synergistic if the summed $AUC_\Delta$ of the two GEM combinations ($\Sigma AUC_\Delta > 10\%$) and if at least one of the two GEM combinations resulted in AUC < 80% and $pEC_{50} > 6$. Only combinations with $R^2 > 0.8$ in both conditions were considered in the synergy analysis.

## Additional cell viability assays

Cell proliferation assays of the combination of GEM and clinical ATR inhibitors in cell lines AsPC-1 ($n = 3$), PANC-1 ($n = 3$), and Suit2-07 ($n = 1$) were performed using the Incucyte live-cell imaging platform (Sartorius). Cells were seeded at 5000 cells/well into 96-well flat bottom plates (Eppendorf) and allowed to attach overnight. For proliferation assays, cells were treated either with ATRi alone (8 doses, half-logarithmic dilution from 3 nM to 10,000 nM), GEM alone (8 doses, half-logarithmic dilution from 1 nM to 3000 nM), or a combination of titrated GEM and three fixed, sub-$EC_{50}$ doses of ATRi (0.2% final DMSO concentration; doses see Dataset EV5, Dataset EV6). The same experimental setup was used to evaluate synergy between ATRi Elimusertib and the four chemodrugs Oxaliplatin, Carboplatin, Paclitaxel, and 5-fluorouracil in AsPC-1 cells ($n = 3$; doses see Dataset EV7). Cells were incubated for 72 h at 37 °C and 5% $CO_2$. Cell confluence readouts at 72 h were either normalized to their initial confluence using the Incucyte software (AsPC-1) or used without prior normalization (all others). Drug treatments were normalized to the median of vehicles per plate (AsPC-1: $n = 5$; all others: $n = 4$) and dose–response curves were fitted to a four-parameter log-logistic model using the R package *drc* (Ritz et al, 2015). Shifts in the $EC_{50}$ between GEM monotherapy and drug pairs were determined to assess combination effects. Each experiment was performed in triplicates, the data was plotted as mean ± standard deviation (s.d.). Significant decrease in $EC_{50}$ compared to GEM monotherapy was determined by pairwise Student's t-test, and p-values were adjusted using the Benjamini–Hochberg procedure (Benjamini and Hochberg, 1995).

For viability assays in PDOs, confluent organoids were removed from medium, and Matrigel was dissolved by adding 200 µl (PDO set 1) or 2 ml (PDO set 2) of Cell Recovery Solution (Corning), and additionally 500 µl of DPBS (PDO set 1 only). Samples were transferred into a falcon and incubated on ice for 30 min, followed by centrifugation at $300 \times g$ at 4 °C for 5 min (after adding 10 ml washing medium for PDO set 2). Subsequently, TrypLE Express Enzyme (Thermo Fisher) was added to cell pellets and incubated for $2 \times 5$ min at 37 °C. The enzymatic reaction was quenched by DPBS (PDO set 1) or washing medium (PDO set 2), and samples were

washed by centrifugation at $300 \times g$ at 4 °C for 5 min. Five hundred cells/well were seeded in 384-well plates (Corning) in 50 μl of PDO medium containing 10% Matrigel. To allow PDO generation, cells were incubated for 4 days at 37 °C and 5% $CO_2$. Then, cells were treated with GEM (1 nM, 10 nM, 100 nM, 1000 nM, 10,000 nM) alone or in combination with three concentrations of ATRi Elimusertib (3 nM, 10 nM, 30 nM) in an additional 10 μl volume of PDO medium and 0.2% DMSO. Experiments were pipetted manually (PDO set 1) or automatically using a TECAN D300e Digital Dispenser (HP; PDO set 2). Treatments were performed in eight replicates, except for PDO PPES045 (quadruplicates). After 72 h, cell viability was measured using the CellTiter-Glo Luminescent Cell Viability Assay (Promega) diluted 1:5 in the final volume of the well. After 20 min of incubation at room temperature under gentle shaking, the acquisition was performed in the microplate reader VantaSTAR (BMG Labtech) using 3000 gain for the amplification measurement of a detected signal (PDO set 1) or the Spark Multimode microplate reader (TECAN) at 800 ms integration time (PDO set 2). Data was normalized to the plate-wise median of vehicles ($n = 4$), and dose–response curves were fitted to a four-parameter log-logistic model with a constrained slope of 1 using the R package *drc*. Significant changes in $EC_{50}$ compared to GEM monotherapy were assessed as described above.

## Chemoproteomic target profiling using Kinobeads

The competition-based drug target profiling assay was performed using Kinobeads ε as described previously (Reinecke et al, 2019). Briefly, 2.5 mg AsPC-1 lysate was incubated with increasing doses of ATRi (1 nM, 3 nM, 10 nM, 30 nM, 100 nM, 300 nM, 1000 nM, 3000 nM, and 30,000 nM) or vehicle for 45 min at 4 °C to allow for target binding. This was followed by incubation with Kinobeads ε (17.5 μl of settled beads) for 30 min at 4 °C to enrich kinases. Unbound proteins from the vehicle experiment were collected and subjected to a second pulldown experiment with fresh beads to assess the degree of protein depletion from the lysate by Kinobeads (Lemeer et al, 2013). Enriched proteins were reduced with 50 mM DTT in 8 M urea, 40 mM Tris-HCl, pH 7.4 for 30 min at room temperature, and alkylated with 55 mM chloroacetamide. After reducing the urea concentration to 1–2 M, proteins were digested with trypsin at 37 °C for 16 h. Subsequently, eluted peptides were desalted using SepPak tC18 μElution plates (Waters), freeze-dried by vacuum centrifugation, and stored at −20 °C.

## LC-MS/MS measurement of Kinobeads experiments

For LC-MS/MS, dried peptides were reconstituted in 0.1% formic acid and analyzed on a Dionex Ultimate3000 nano HPLC (Thermo Scientific) coupled to an Orbitrap Fusion Tribrid mass spectrometer (Thermo Scientific), which was run in data-dependent mode. Peptides were first delivered to a trap column (100 μm × 2 cm, packed in-house with Reprosil-Gold $C_{18}$ ODS-3 5 μm resin, Dr. Maisch, Ammerbuch) and subsequently washed with solvent A0 (0.1% formic acid in HPLC grade water) at 5 μl/min for 10 min. Then, peptides were separated on an analytical column (75 μm × 40 cm, packed in house with Reprosil-Gold $C_{18}$ 3 μm resin, Dr. Maisch, Ammerbuch) at 300 nl/min using a 52 min gradient ranging from 4–32% solvent B (0.1% formic acid, 5% DMSO in acetonitrile) in solvent A1 (0.1% formic acid, 5% DMSO in HPLC grade water). MS1 spectra were acquired in the Orbitrap at a resolution of 60,000 (at $m/z$ 200) over a scan range of 360–1300 $m/z$

using a maximum injection time of 50 ms and an automatic gain control (AGC) target value of $4e^5$ (normalized AGC target 100%). For MS2, up to 12 peptide precursors were isolated (isolation width of 1.2 Th) and subjected to HCD fragmentation using 30% normalized collision energy. Fragmented precursor ions were analyzed in the Orbitrap at a resolution of 30,000, with a maximum injection time of 75 ms and an AGC target value of $1e^5$ (normalized AGC target 200%). The duration of dynamic exclusion was set to 30 s.

## Data processing of Kinobeads experiments

For protein and peptide identification and quantification, raw data was searched against the human reference proteome including isoforms (downloaded from UniProt on March 16, 2021) using MaxQuant (Cox and Mann, 2008) (v1.6.12.0). Oxidized methionine and N-terminal acetylation were set as variable modification, and cysteine carbamidomethylation was set as fixed modification. Label-free quantification (LFQ) and match-between-runs were enabled. All searches were performed with 1% PSM and protein FDR.

Proteins were filtered by potential contaminants, reversed hits and proteins identified only by site. Residual binding of proteins was calculated from LFQ intensities as the ratio between drug doses and vehicle. Data points were fitted using CurveCurator (Bayer et al, 2023) (default parameters). Kinases were considered a potential target if curve goodness of fit $R^2 > 0.7$, curve fold change < 0.5, curve slope > 0.2, and $pEC_{50} > 6$. Only kinases quantified with at least 4 unique peptides in the vehicle experiment were considered. Potential targets were further curated through manual inspection of dose-dependent reduction in MS/MS and unique peptide counts. To calculate apparent dissociation constants ($K_D^{app}$), the $EC_{50}$ value of each protein was multiplied with a correction factor (the protein intensity ratio of the two subsequent pulldown experiments (Lemeer et al, 2013), mean of all four experiments).

## Drug treatment for (phospho-)proteomic readout

For proteomic drug-perturbation experiments, cell lines AsPC-1, MiaPaCa-2, PANC-1, PaTu-8988-T, Dan-G, PSN-1, and Capan-1 were seeded in 10 $cm^2$ dishes and grown to 80% confluence within 48 h. Cell culture medium was replaced 24 h after seeding. For decryptM, AsPC-1 cells were first pre-incubated with vehicle or 1 μM GEM for 3 h. Then, either vehicle or 9 increasing concentrations of ATRi (1 nM, 3 nM, 10 nM, 30 nM, 100 nM, 300 nM, 1000 nM, 3000 nM, and 10,000 nM) were added to the cells and incubated for one more hour. For all other cell lines, cells were pre-incubated with vehicle or 1 μM GEM for 3 h and subsequently incubated with either vehicle or 1 μM ATRi for one more hour (in quadruplicates). For time-resolved experiments, AsPC-1 cells were treated with either 1 μM GEM, 1 μM ATRi, or vehicle for eight different time points (15 min, 30 min, 1 h, 2 h, 4 h, 8 h, 12 h, 24 h). For time point 0, AsPC-1 cells were treated with a vehicle and lysed immediately. The final DMSO concentration in all experiments was 0.2%.

## Sample preparation of (phospho-)proteomic experiments

### SDS cell lysis and protein quantification

Cells were lysed as previously described (Zecha et al, 2020) with minor differences. Briefly, cells were washed twice with PBS and lysed in SDS lysis buffer (2% SDS in 40 mM Tris-HCl, pH 7.6), followed by

sonication for 10 min (30 s on/30 s off) using a Bioruptor (Diagenode). For DNA hydrolysis, samples were boiled at 95 °C for 10 min and incubated with 2% trifluoroacetic acid (TFA) for 1 min, before the reaction was quenched using 4% N-methylmorpholine. Lysate was cleared by centrifugation at $11,000 \times g$ for 5 min and protein concentration was determined using the Pierce BCA Protein Assay Kit (Thermo Scientific).

Sample clean-up and digest was performed using the SP3 method on an automated Bravo liquid handling platform (Agilent Technologies, CA, USA) as previously described (Hughes et al, 2019; Zecha et al, 2020) with minor modifications. In brief, 200 µg of protein lysate was mixed with 1000 µg carboxylate beads (1:1 mix of magnetic SpeedBeads 45152105050250 and 65152105050250, Cytiva) in a 96-well plate. Using 70% ethanol, samples were then precipitated on beads, followed by washing thrice with 80% ethanol and once with 100% acetonitrile. Subsequently, proteins were reduced and alkylated using 10 mM TCEP, 50 mM chloroacetamide, and 2 mM $CaCl_2$ in 100 mM EPPS/NaOH, pH 8.5 for 1 h at 37 °C, followed by tryptic digestion at 37 °C for 16 h. Peptides were acidified with 1% TFA and desalted on Chromabond HLB plates (30 µm particle size, 10 mg capacity, Machery-Nagel), using 0.1% TFA for equilibration and sample loading, and 0.1% TFA in 70% acetonitrile for peptide elution. Recovered peptides were freeze-dried by vacuum-centrifugation and stored at −20 °C.

### TMT labeling

Depending on the experiment, samples were either labeled with TMT-11plex (decryptM), TMT-10plex (non-decryptM, quadruplicates), or TMTpro reagent (time-series; all Thermo Scientific). Therefore, dried peptides were reconstituted in 15 µl 100 mM EPPS, pH 8.5 and each treatment condition was labeled with one channel of the respective TMT reagent (5 µl of 25 µg/µl reagent, Appendix Tables S4–S6). The reaction was incubated for 1 h at 23 °C and subsequently quenched with 0.4% hydroxylamine. Then, samples were pooled, acidified with 1% formic acid, and freeze-dried by vacuum-centrifugation. Labeled peptide pools were desalted on $C_{18}$ Sep-Pak cartridges (37–55 µm particle size, 50 mg capacity, Waters), using 0.1% formic acid as equilibration solvent, and 0.1% formic acid in 60% acetonitrile was used as elution solvent. Eluted peptides were freeze-dried and stored at −20 °C until further use.

### Offline basic reversed-phase fractionation

Basic reversed-phase (bRP) fractionation of peptides was performed on a Vanquish HPLC (Thermo Scientific). In short, dried peptides were reconstituted in 200 µl 25 mM ammonium bicarbonate, pH 8.0 and directly injected onto a Waters BEH130 XBridge $C_{18}$ column (3.5 µm, 4.6 × 250 mm). At a flow rate of 1000 µl/min, peptides were eluted using a 60 min gradient ranging from 7% to 45% acetonitrile in the constant presence of 2.5 mM ammonium bicarbonate, pH 8.0. Using an automated fraction collector, 96 fractions were collected between min 7 and 55 after injection. Fractions were acidified with 1% formic acid and pooled to 48 fractions, before they were freeze-dried and stored at −20 °C.

### Phosphopeptide enrichment using IMAC

Immobilized metal ion affinity chromatography (IMAC) enrichment of phosphorylated peptides was performed on the Bravo automated liquid handling platform using AssayMAP Fe(III)-NTA

cartridges (Agilent). In brief, the 48 dried peptide fractions were reconstituted in 0.1% TFA in 80% acetonitrile and further pooled to 12 fractions (200 µl final volume per well). The phosphopeptide enrichment protocol embedded in the Agilent AssayMAP Bravo Protein Sample Prep Workbench v2.0 software was run. Thereby, the cartridges were first primed with 150 µl 0.1% TFA in acetonitrile at 300 µl/min, and equilibrated with 150 µl 0.1% TFA in 80% acetonitrile at 10 µl/min. Then, pooled fractions were loaded at 5 µl/min and further washed three times with 150 µl 0.1% TFA in 80% acetonitrile at 50 µl/min. Enriched peptides were eluted with 60 µl 1% ammonium hydroxide at 5 µl/min. The collected eluates were acidified with 1% formic acid, freeze-dried, and stored at −20 °C until LC-MS/MS measurement.

## LC-MS/MS measurement of (phospho-) proteomic samples

Phosphorylated TMT-labeled peptides were measured on a Fusion Lumos Tribrid mass spectrometer (Thermo Scientific) coupled to a Dionex UltiMate 3000 RSLCnano System (Thermo Scientific) in data-dependent mode using MS3-quantification. In brief, dried peptides were reconstituted in 50 mM sodium citrate buffer injected onto a trap column (100 µm × 2 cm, packed in-house with Reprosil-Gold $C_{18}$ ODS-3 5 µm resin, Dr. Maisch, Ammerbuch) and subsequently washed with solvent A0 (0.1% formic acid in HPLC grade water) at a flow rate of 5 µl/min for 10 min. Then, peptides were separated on an analytical column (75 µm × 48 cm, packed in house with Reprosil-Gold $C_{18}$ 3 µm resin, Dr. Maisch, Ammerbuch) at a flow rate of 300 nl/min using an 80 min, two-step gradient. For minutes 0–65, the gradient ranged from 4–22.5% solvent B (0.1% formic acid, 5% DMSO in acetonitrile) in solvent A1 (0.1% formic acid, 5% DMSO in HPLC grade water), and from 22.5–32% solvent B for minutes 65–80. Peptides were ionized using a nano-source with 2.1 kV spray voltage. MS1 spectra were acquired in the Orbitrap at a resolution of 60,000 (at 200 $m/z$) over a scan range of 360 to 1800 $m/z$, using an AGC target value of $4e^5$ and a maximum injection time of 50 ms. The cycle time between consecutive MS1 scans was 3 s. Precursor ions were isolated using a quadrupole isolation window of 0.7 Th, and subjected to CID fragmentation in the linear ion trap using 35% normalized collision energy, with multistage activation enabled. MS spectra were acquired in the Orbitrap at a resolution of 30,000 over an auto scan range, using an AGC target of $5e^4$ and a maximum injection time of 60 ms, with the inject-beyond feature enabled. The duration of dynamic exclusion was set to 90 s. Using a charge state-dependent MS3 quadrupole isolation window of 1.2 Th (z = 2), 0.9 Th (z = 3), 0.7 Th (z = 4–6), a new batch of TMT reporter ions was isolated for a consecutive MS3 scan. Using synchronous precursor selection, the top 10 fragment ions of the MS2 scans were isolated and subjected to HCD fragmentation in the linear ion trap using 55% normalized collision energy. MS3 spectra were acquired in the Orbitrap at a resolution of 50,000 over a scan range of 100 to 1000 Th, using an AGC target of $1e^5$ and a maximum injection time of 120 ms.

LC-MS/MS measurement of non-enriched, TMT-labeled peptides from temporal drug perturbation was carried out on an Orbitrap Fusion Lumos Tribrid instrument (Thermo Scientific) coupled to a micro-flow LC system built by combining a modified Vanquish pump and the autosampler of the Dionex UltiMate 3000 nano HPLC System

(Thermo Scientific). Dried peptides were reconstituted in 0.1% formic acid and loaded directly onto an Acclaim PepMap 100 $C_{18}$ column (2 μm particle size, 1 mm × 150 mm, Thermo Scientific) heated at 55 °C. Samples were separated at a flow rate of 50 μl/min using an 25 min linear gradient of 4–32% solvent B (0.1% formic acid, 3% DMSO in acetonitrile) in solvent A (0.1% formic acid, 3% DMSO in HPLC grade water). Peptides were ionized with an electrospray voltage of 3.5 kV, a capillary temperature of 325 °C, and a vaporizer temperature of 125 °C. Flow rates of sheath, aux, and sweep gas were kept at 32, 5, and 0, respectively. MS1 spectra were acquired in the Orbitrap at a resolution of 60,000 using a maximum injection time of 50 ms and an AGC target value of $4e^5$. The cycle time between MS1 scans was 1.2 s, with a dynamic exclusion duration of 50 s. Precursor ions were isolated with a quadrupole isolation window of 0.6 Th and fragmented by HCD using a normalized collision energy of 32%. MS2 spectra were collected in the linear ion trap in rapid scan mode using a maximum injection time of 40 ms and an AGC target value of $1.2e^4$ (auto scan range). Of the MS2 scans, the top 8 fragment ions were isolated for MS3 using synchronous precursor selection with isolation windows of 1.2 Th and subjected to HCD fragmentation in the linear ion trap at 55% normalized collision energy. MS3 spectra were acquired in the Orbitrap at a resolution of 50,000 over a scan range of 100 to 1000 *m/z*, with a maximum injection time of 86 ms and an AGC target value of $1e^5$.

### Data processing of drug perturbation experiments

For protein and peptide identification and quantification, phospho-proteomic raw data was searched against the human reference proteome (downloaded from UniProt on March 16, 2021) using MaxQuant (v1.6.12.0). Oxidized methionine, N-terminal acetylation, and phosphorylation (STY) were set as variable modification, and cysteine carbamidomethylation was set as fixed modification. All other search parameters were kept as default, except for protein FDR which was set to 1. The resulting evidence.txt and msms.txt files were subjected to SIMSI-Transfer (Hamood et al, 2022) (v0.5.0) using a p10 threshold. Potential contaminants and reversed hits were removed, and only peptides with unambiguously assigned phosphorylation sites with a localization probability of > 0.75 were retained. To acquire site-level quantification values, the corrected reporter ion intensities of peptides containing the same phospho-site(s) were summed up. Median centering of corrected TMT reporter channel intensities was performed. For decryptM in AsPC-1 cells, channels 1–10 (decryptM in GEM-treated cells) were submitted to CurveCurator (v0.4.1) for curve fitting and classification of significantly regulated dose–response relationships based on relevance score, using a 2D threshold (α-limit: 0.05 and $\log_2$ fold change limit: 0.45; default parameters (Bayer et al, 2023)). The resulting list of regulated curves was additionally filtered for $pEC_{50} > 5$. Channel 11 served as a vehicle for the chemodrug treatment and was not subjected to curve fitting. Instead, row-wise normalization was performed across all four TMT-batches using the medians of channels 10 and 11. Intensities were then $\log_2$ transformed and submitted to two-sided t-test analysis in Perseus (Tyanova et al, 2016), with at least three valid values per group (GEM or vehicle) and multiple testing correction using the Benjamini–Hochberg procedure (Benjamini and Hochberg, 1995). Regulation was defined significant when adjusted *p*-values (*q*-values) < 0.01 and absolute $\log_2$ fold changes > 1. Phosphoproteomic

experiments in six additional cell lines (not dose-resolved) were median-centered within each TMT-batch, prior to row-wise normalization across all TMT-batches using a common channel filled with AsPC-1 lysate (Appendix Table S5). Two-sided t-test analysis and multiple hypothesis testing correction was performed in R using the same parameters and thresholds as described above.

Peptide and protein identification and quantification of time-series raw data was performed with MaxQuant (v2.1.3.0) using the same sequence information as above, with N-terminal acetylation and oxidized methionine as variable modifications and cysteine carbamidomethylation as fixed modification. All other search parameters were set as default. The resulting proteingroups.txt file was filtered to remove potential contaminants and reversed hits. Corrected TMT reporter channel intensities were median-centered, and ratios were calculated between treatments and their respective time-point controls (Appendix Table S6). Proteins were considered conversely regulated if absolute $\log_2$ fold change > 2 by at least one drug and oppositely affected by both drugs (up/down) at 24 h.

## Data availability

The datasets and computer code produced in this study are available in the following databases: Mass spectrometry proteomics data: ProteomeXchange Consortium (Deutsch et al, 2023) via the MassIVE partner repository, PXD050707. Processed dose–response data and plots: Zenodo, 10.5281/zenodo.10792252 (https://zenodo.org/records/10792252). Proteomic decryptM data for inter-active exploration: ProteomicsDB, PRDB004510. Computational code for proteomic data analysis and plotting: Github repository (https://github.com/kusterlab/Gem_synergy).

The source data of this paper are collected in the following database record: biostudies:S-SCDT-10_1038-S44320-025-00085-6.

## Peer review information

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

## Acknowledgements

We thank Prof Kirsten Lauber from Radiation/Oncology, LMU for providing cell lines Suit2-07, Colo-357 and HPDE, and the Chemical Biology Core Facility at EMBL Heidelberg for providing expertise and support for drug screen experiments. Further, we thank Dr Thomas Herold from the Institute of Pathology at the University Hospital Essen for mutational analysis of pancreatic patient-derived organoids (PDO set 2). HS receives support from the University Medicine Essen Clinician Scientist Academy (UMEA) in the framework of the DFG Clinician Scientist Program (DFG; FU 356/12-2). JTS is grateful for support from the German Cancer Consortium (DKTK) and the German Federal Ministry of Education and Research (BMBF; 01KD2206A/ SATURN3). This work was funded by the German Research Foundation (DFG; SFB1321, grant number 329628492), the German Federal Ministry of Education and Research (BMBF; DROP2AI, grant number 031L0305A), and the European Research Council (ERC; TOPAS, grant number 833710). Parts of the figures were created with BioRender.com.

## Author contributions

**Stefanie Höfer**: Conceptualization; Data curation; Formal analysis; Supervision; Validation; Investigation; Visualization; Methodology; Writing—original draft; Writing—review and editing. **Larissa Frasch**: Formal analysis; Investigation; Methodology. **Sarah Brajkovic**: Formal analysis; Validation. **Kerstin Putzker**: Formal analysis; Methodology. **Joe Lewis**: Resources; Methodology. **Hendrik Schürmann**: Formal analysis; Validation; Methodology. **Valentina Leone**: Formal analysis; Validation; Methodology. **Amirhossein Sakhteman**: Data curation; Visualization. **Matthew The**: Resources; Software. **Florian P Bayer**: Software; Methodology. **Julian Müller**: Software; Methodology. **Firas Hamood**: Software; Methodology. **Jens T Siveke**: Conceptualization; Resources; Methodology. **Maximilian Reichert**: Conceptualization; Resources; Methodology. **Bernhard Kuster**: Conceptualization; Resources; Formal analysis; Supervision; Funding acquisition; Methodology; Writing—original draft; Project administration; Writing—review and editing.

Source data underlying figure panels in this paper may have individual authorship assigned. Where available, figure panel/source data authorship is listed in the following database record: biostudies:S-SCDT-10_1038-S44320-025-00085-6.

## Funding

## Disclosure and competing interests statement

BK is co-founder and shareholder of OmicScouts and MSAID. He has no operational role in either company. JTS receives honoraria as consultant or for continuing medical education presentations from AstraZeneca, Bayer, Boehringer Ingelheim, Bristol-Myers Squibb, Immunocore, MSD Sharp Dohme, Novartis, Roche/Genentech, and Servier. His institution receives research funding from Abalos Therapeutics, AstraZeneca, Boehringer Ingelheim, Bristol-Myers Squibb, Celgene, Eisbach Bio, and Roche/Genentech; he holds ownership in FAPI Holding (< 3%); all outside the submitted work. HS received travel support from Servier, with no financial benefit and outside the submitted work. The remaining authors declare no competing interests.

# Expanded View Figures

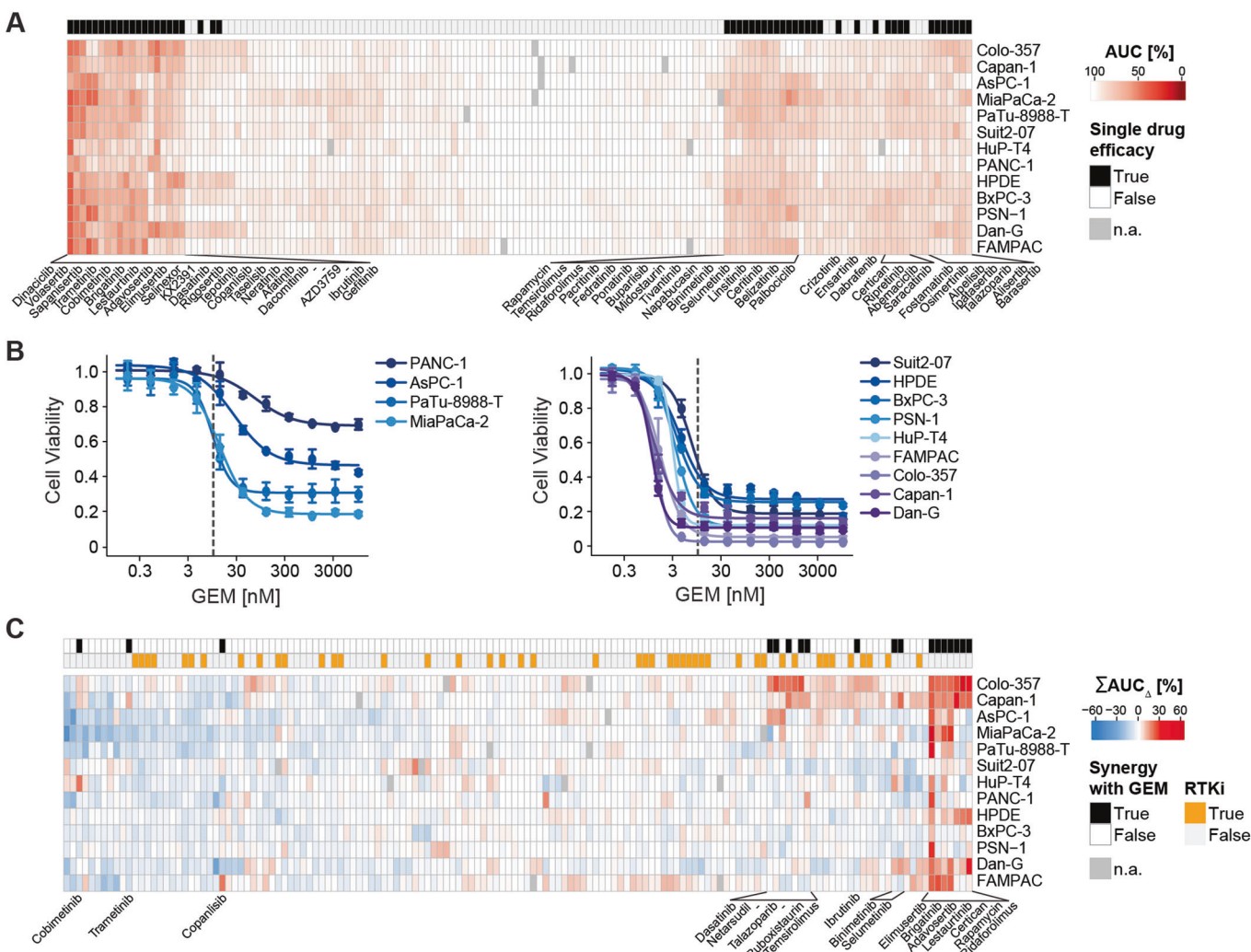

**Figure EV1. Single drug and combination screening of 146 targeted inhibitors and GEM in 13 PDAC cell lines.**

(A) Area-under-curve (AUC, in %) of single drug treatments for all 146 inhibitors across 13 PDAC cell lines. A lower AUC indicates greater efficacy. Drugs showing efficacy in at least one cell line are annotated in black, and labeled with text. Missing data (n.a.) is indicated in gray. (B) Cell viability upon treatment with increasing doses of GEM for all cell lines, relative to vehicle. Data are presented as mean values, with error bars representing the ± s.d. of duplicates ($n = 2$). Dashed line indicates an $EC_{50}$ threshold of 10 nM, which was used to separate less sensitive cell lines (left) from more sensitive cell lines (right). (C) Summed shift in AUC ($\Sigma AUC_\Delta$, in %) upon combination with GEM for all 146 library drugs across all cell lines. A higher $\Sigma AUC_\Delta$ indicates greater synergy. Drugs showing synergy in at least one cell line are annotated in black, and labeled with text. Inhibitors of receptor tyrosine kinases (RTKi) are annotated in orange. Missing data (n.a.) is indicated in gray.

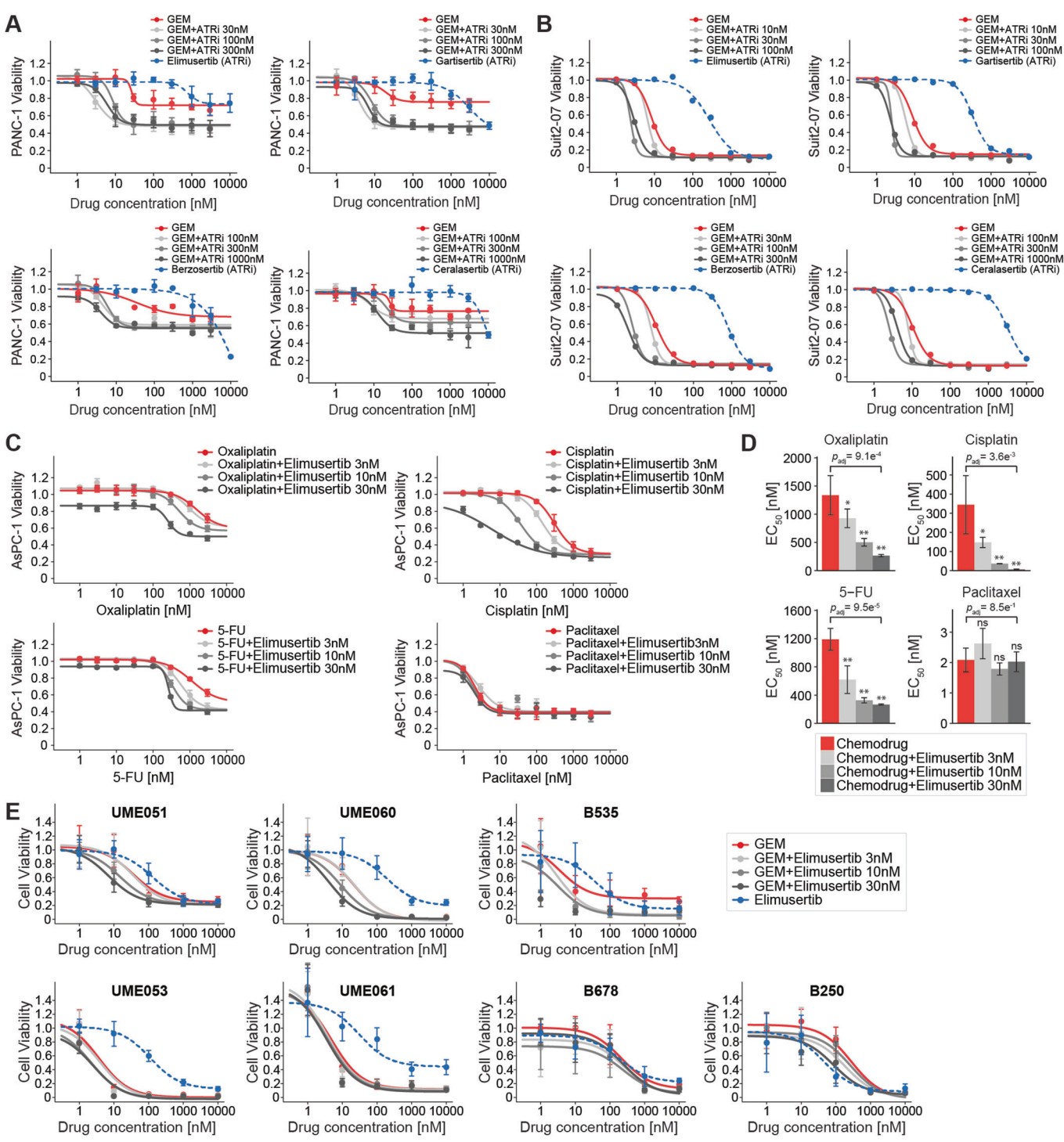

**Figure EV2. Additional viability assays with ATRi in PDAC cell lines and PDOs.**

(A, B) Cell viability of PANC-1 (A) and Suit2-07 (B) after treatment with GEM alone (red), GEM in combination with three sub-$EC_{50}$ doses of ATR inhibitor (shades of gray), or ATR inhibitor alone (blue dotted) relative to vehicle. (C) AsPC-1 viability after treatment with four different chemodrugs alone (red) or chemodrugs in combination with three sub-$EC_{50}$ doses of Elimusertib (shades of gray) relative to vehicle. (D) $EC_{50}$ of four different chemodrugs alone (red) and chemodrugs in combination with three sub-$EC_{50}$ doses of Elimusertib (shades of gray). Asterisks show the significance level from a Student's t-test against GEM monotherapy (p-values were adjusted using the Benjamini–Hochberg procedure; $^*p_{adj} < 0.05$; $^{**}p_{adj} < 0.01$; ns: non-significant). Adjusted p-values are shown only for the most significant combinations; for all others, refer to Dataset EV7. (E) Cell viability of seven PDOs treated with GEM alone (red), GEM in combination with three sub-$EC_{50}$ doses of Elimusertib (shades of gray), or Elimusertib alone (blue dotted) relative to vehicle. Data information: Data are presented as mean values, with error bars representing the ± s.d. of triplicates (A, C, and D; $n = 3$) or eight replicates (E; $n = 8$). Data shown in (B) represent a single experiment ($n = 1$).

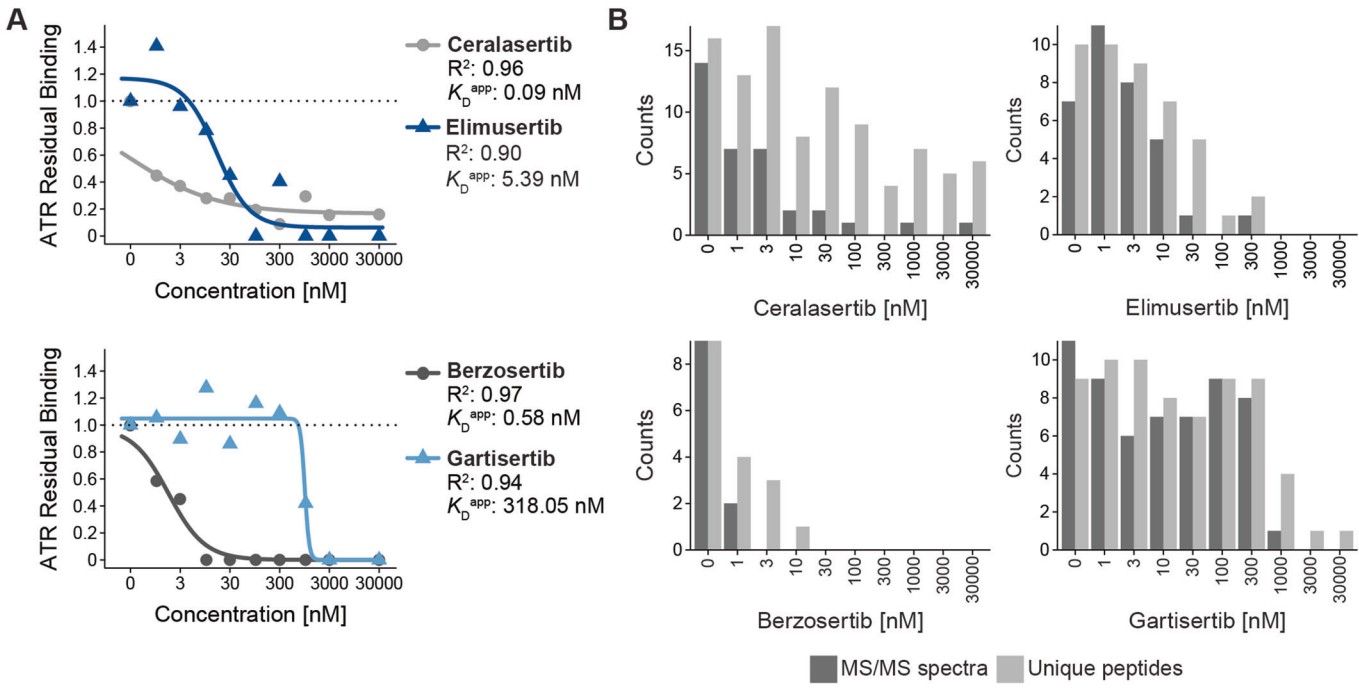

**Figure EV3.  Binding of ATR kinase by clinical inhibitors.**

(**A**) Residual binding of ATR on Kinobeads upon increasing doses of Elimusertib and Ceralasertib (top), and Berzosertib and Gartisertib (bottom), based on label-free quantification (LFQ) intensities. Curve fit ($R^2$) and apparent affinity constants ($K_D^{app}$) are given in the legend. (**B**) Dose-dependent reduction in MS/MS spectra and unique peptides of ATR kinase in pulldown experiment with increasing concentrations of Elimusertib and Ceralasertib (top), and Berzosertib and Gartisertib (bottom).

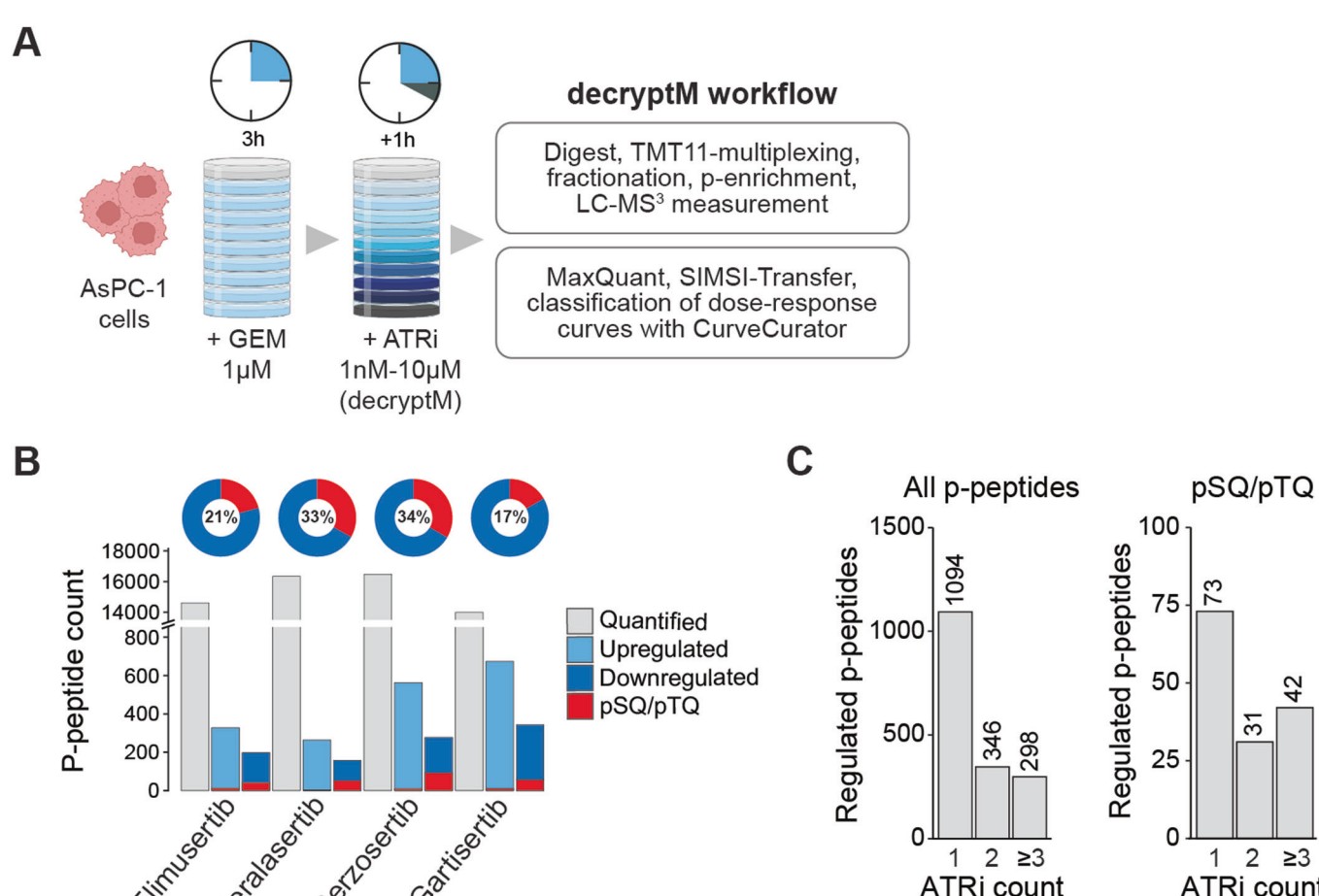

**Figure EV4.  Phosphoproteomic decryptM workflow to study ATR inhibition in DNA-damaged cells.**

(A) Schematic workflow of decryptM experiments with four clinical ATR inhibitors in DNA-damaged AsPC-1 cells (pre-incubated with GEM). Seeded AsPC-1 cells were incubated with 1 μM GEM for 3 h, followed by nine doses of ATR inhibitor for one additional hour. TMT-labeled, fractionated and phospho-enriched peptides were measured by LC-MS[3]. After data processing with MaxQuant and SIMSI-Transfer, dose–response data was analyzed using CurveCurator. (B) Number of quantified (gray), upregulated (light blue), or downregulated (dark blue) phosphorylated peptides in the four decryptM experiments. Peptides containing the pSQ/pTQ motif are highlighted in red, and the numbers in pie charts indicate the fraction of pSQ/pTQ motif-containing peptides within the down-regulated phosphoproteome. (C) Count of all p-peptides (left) or pSQ/pTQ motif-containing peptides (right) regulated by one, two, or at least three ATR inhibitors in decryptM experiments.

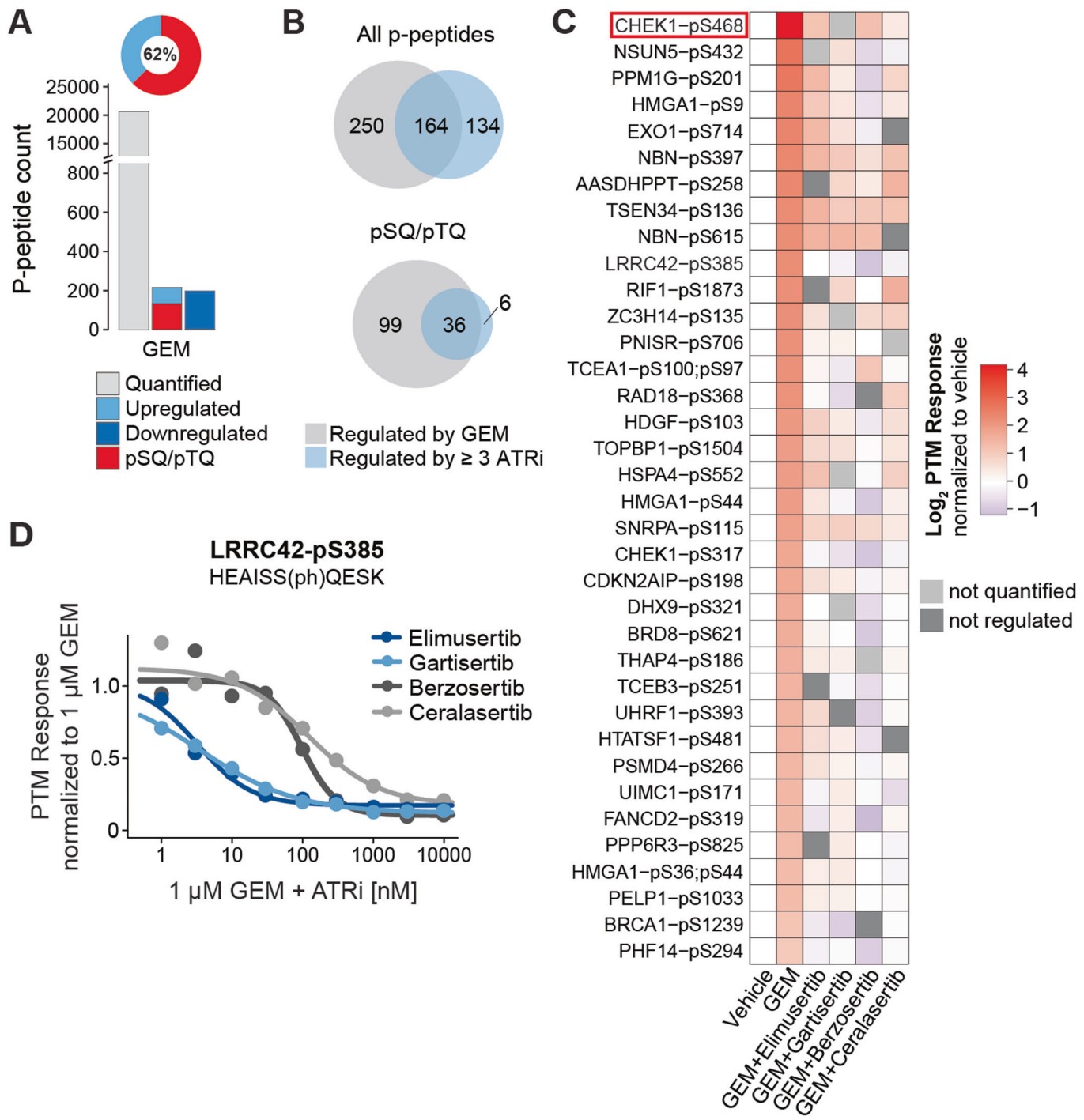

**Figure EV5.  36 GEM-induced pSQ/pTQ sites are counter-regulated by ATRi.**

(**A**) Barplot showing the number of all quantified (gray), upregulated (light blue), or downregulated (dark blue) phosphorylated peptides upon treatment of AsPC-1 cells with 1 μM GEM for 4 h (*n* = 4). Peptides containing the pSQ/pTQ motif are highlighted in red, and numbers in pie charts indicate the fraction of pSQ/pTQ peptides within the upregulated phosphoproteome. (**B**) Overlap in regulated phosphorylated peptides between GEM and at least three out of four ATR inhibitors. Left: all p-peptides, right: pSQ/pTQ motif-containing peptides. (**C**) Heatmap of phosphorylation sites induced by GEM and inhibited by at least three of the four ATR inhibitors. Log₂ fold changes in phosphorylation are shown relative to vehicle (no drug). Light gray indicates missing values (not quantified), and dark gray indicates quantified but not significantly regulated peptides. (**D**) Dose-dependent regulation of LRRC42-pS385 by the four ATR inhibitors in DNA-damaged cells. PTM response was normalized to 1 μM GEM.

