## [Peer Review File · Molecular Systems Biology]

Gemcitabine and ATR inhibitors synergize to kill PDAC cells by blocking DNA damage response

Stefanie Höfer, Larissa Frasch, Sarah Brajkovic, Kerstin Putzker, Joe Lewis, Hendrik Schürmann, Valentina Leone, Amirhossein Sakhteman, Matthew The, Florian Bayer, Julian Müller, Firas Hamood, Jens Siveke, Maximilian Reichert, and Bernhard Küster

Corresponding author(s): Bernhard Küster (kuster@tum.de)

Review Timeline:

Submission Date:	2nd Apr 24
Editorial Decision:	21st May 24
Revision Received:	17th Oct 24
Editorial Decision:	26th Nov 24
Revision Received:	22nd Dec 24
Accepted:	3rd Jan 25

Editor: Poonam Bheda

Transaction Report:

21st May 2024

Manuscript Number: MSB-2024-12345

Title: Gemcitabine and ATR inhibitors synergize to kill PDAC cells by blocking DNA damage response

Dear Bernhard,

Thank you again for submitting your work to Molecular Systems Biology. We have now heard back from the three reviewers who agreed to evaluate your study. As you will see below the reviewers acknowledge that the study is a relevant contribution to the field. However, they raise several concerns, which we would invite you to address in a revision.

I think that the reviewers' recommendations are rather clear and I therefore see no need to repeat the comments listed below. All issues raised by the reviewers would need to be satisfactorily addressed. As you may already know, our editorial policy allows in principle a single round of major revision. It is therefore essential to provide responses to the reviewers' comments that are as complete as possible. If you have any questions of if would like to discuss your revision plan with me, please feel free to get in touch.

On a more editorial level, we would ask you to address the following points:

- Please provide a .doc version of the manuscript text (including legends for main Figures and EV Figures) and individual production quality figure files for the main Figures and EV Figures (one file per figure).
- Please include 5 keywords.
- We have replaced Supplementary Information by the Expanded View (EV format). In this case all additional figures can be provided as EV Figures. Please provide one file per EV Figure. Their legends should be included in the manuscript text. For detailed instructions regarding expanded view please refer to our Author Guidelines: .
- EV Tables that are long (i.e. longer than one page) and/or complex should be provided as EV Datasets. Please include a description of the EV Dataset (or EV Table) in a separate sheet in the xls file. One file should be provided per EV Table/Dataset.
- Please provide a "standfirst text" summarizing the study in one or two sentences (approximately 250 characters), three to four "bullet points" highlighting the main findings and a "synopsis image" (exactly 550px width and max 400px height, jpeg or png format) to highlight the paper on our homepage.
- All Materials and Methods need to be described in the main text. We would encourage you to use 'Structured Methods', our new Materials and Methods format. According to this format, the Material and Methods section should include a Reagents and Tools Table (listing key reagents, experimental models, software and relevant equipment and including their sources and relevant identifiers) followed by a Methods and Protocols section in which we encourage the authors to describe their methods using a step-by-step protocol format with bullet points, to facilitate the adoption of the methodologies across labs. More information on how to adhere to this format as well as downloadable templates (.doc or .xls) for the Reagents and Tools Table can be found in our author guidelines: . An example of a Method paper with Structured Methods can be found here:
- Please include a Data availability section describing how the data and code have been made available. This section needs to be formatted according to the example below:
The datasets and computer code produced in this study are available in the following databases:
 - Chip-Seq data: Gene Expression Omnibus GSE46748 (<https://www.ncbi.nlm.nih.gov/geo/query/acc.cgi?acc=GSE46748>)
 - Modeling computer scripts: GitHub (<https://github.com/SysBioChalmers/GECKO/releases/tag/v1.0>)
 - [data type]: [full name of the resource] [accession number/identifier] ([doi or URL or identifiers.org/DATABASE:ACCESSION])
- For data quantification: please specify the name of the statistical test used to generate error bars and P values, the number (n) of independent experiments (specify technical or biological replicates) underlying each data point and the test used to calculate p-values in each figure legend. The figure legends should contain a basic description of n, P and the test applied. Graphs must include a description of the bars and the error bars (s.d., s.e.m.).
- Please include a "Disclosure & Competing Interests Statement" in the main text.
- The References should be formatted according to the Molecular Systems Biology reference style (i.e., ordered alphabetically and listing the first 10 authors followed by et al).
- When you resubmit your manuscript, please download our CHECKLIST (<https://bit.ly/EMBOPressAuthorChecklist>) and include

the completed form in your submission.

Please note that the Author Checklist will be published alongside the paper as part of the transparent process (<https://www.embopress.org/page/journal/17444292/authorguide#transparentprocess>).

If you feel you can satisfactorily deal with these points and those listed by the referees, you may wish to submit a revised version of your manuscript. Please attach a covering letter giving details of the way in which you have handled each of the points raised by the referees. A revised manuscript will be once again subject to review and you probably understand that we can give you no guarantee at this stage that the eventual outcome will be favorable.

Kind regards,

Maria

Maria Polychronidou, PhD
Senior Editor
Molecular Systems Biology

We realize that it is difficult to revise to a specific deadline. In the interest of protecting the conceptual advance provided by the work, we recommend a revision within 3 months (19th Aug 2024). Please discuss the revision progress ahead of this time with the editor if you require more time to complete the revisions. Use the link below to submit your revision:

Link Unavailable

IMPORTANT:

See also figure legend guidelines: <https://www.embopress.org/page/journal/17444292/authorguide#figureformat>

- Please note that corresponding authors are required to supply an ORCID ID for their name upon submission of a revised manuscript (EMBO Press signed a joint statement to encourage ORCID adoption).

(<https://www.embopress.org/page/journal/17444292/authorguide#editorialprocess>)

Currently, our records indicate that the ORCID for your account is 0000-0002-9094-1677.

Link Not Available

***** PLEASE NOTE ***** As part of the EMBO Press transparent editorial process initiative (see our Editorial at <https://dx.doi.org/10.1038/msb.2010.72>), Molecular Systems Biology publishes online a Review Process File with each accepted manuscripts. This file will be published in conjunction with your paper and will include the anonymous referee reports, your point-by-point response and all pertinent correspondence relating to the manuscript. If you do NOT want this File to be published, please inform the editorial office at msb@embo.org within 14 days upon receipt of the present letter.

Reviewer #1:

In the paper "Gemcitabine and ATR inhibitors synergize to kill PDAC cells by blocking DNA damage response" Höfer et al. perform initially a phenotypic screen followed by a proteomic screen to test synergy of the DNA-damage agent Gemcitabine (GEM) with kinase inhibitors in 13 human pancreatic adenocarcinoma cell lines. They found that generally ATR inhibitors (ATRi) synergize with GEM across the majority of cell lines. Furthermore, using dose-dependent phosphoproteomics on cells that were treated with GEM alone compared to the combination treatment with (ATRi), the authors could provide hypotheses for the synergistic effect and suggest possible phosphorylation sites that might be critical for the synergy.

Overall, I think this is an elegant, comprehensive, and well-performed study to test drug synergies across multiple cell lines and many drugs. While the overall outcome on the synergies is not entirely novel as GEM and ATRi synergies have been reported before, the novelty of the study in my opinion lies in providing (1) a general solution for in vitro testing and prioritizing drug synergies, which can be expanded to other targets and (2) a data resource that can be mined to derive mechanistic hypothesis for more targeted follow up studies.

There are a few additional experiments and comments that the authors should address in the current version before publication.

Major comments

- While the panel of 4 ATRi in combination with GEM are only tested in AsPC-1 cells, the question remains whether the inhibition of ATR has a broad synergistic effect with GEM across the other cell lines tested or whether the broad synergistic effect is specific for Elimuserib?
- Based on the DecryptM phosphoproteomics approach the authors can pinpoint specific phosphorylation sites that might be indicative of the synergistic effect of GEM and ATRi, among these are sites that have not previously been linked to DNA damage. While it would be very exciting to test these sites functionally to validate their effect on DNA damage, these experiments would be beyond the scope of this study. However, the authors suggest these sites could serve as biomarkers for drug response for the combination of GEM and ATRi. One important aspect of a drug response marker is that these markers show a consistent effect that is not specific for a certain drug or cell lines. While the authors provide data that these phosphorylation site changes are consistent across different ATRi inhibitors in one cell line, it would be important to assess whether these phosphorylation sites are also consistently responding across the different cell lines. Therefore, I would suggest an experiment to test the consistency of the phosphorylation site changes across the panel of cell lines for the GEM/Elimusertib combination. This would not only strengthen the claim for certain phosphosites as response markers, which I believe represents a novelty of the study, but would in my opinion also be within the scope of this study.

Minor comments

- It would be beneficial for the reader if some additional description for the phenotypic drug screen would be provided in the first section of the manuscript. Currently, it is not apparent from the text what the phenotypic readout for the drug screen actually is, only by looking at the figure readers will know that there is a cell viability readout.

Reviewer #2:

This is an important and well-written piece of work. I have only few comments:

MAJOR CONCERNS

Gemcitabine is not really the mainstay of systemic treatment of PDAC anymore. The authors should provide a stronger rationale why gemcitabine was the backbone for testing combinations.

Related to that; can the authors demonstrate that the synergy is specific to gemcitabine and not the other commonly used cytotoxics (paclitaxel, 5-FU, oxaliplatin, irinotecan/SN-38)?

HPDE cells are not normal or healthy. The authors can remove the caveat at the bottom of page 9 but I would also advise to test the combination in organoids. In this system, truly healthy cells can be compared to tumour-derived organoids. This would also add to the translational potential (in lieu of mouse experiments that I do not recommend).

MINOR COMMENTS

I feel that a lot of what is now in the Extended Data could actually be shown in the main figures as well. I do not think it would add too much bulk.

What type of phosphoproteomics was performed is not immediately clear in the Results or Figure Legend. Please add.

Figure 2 has an L-shaped line around the figure.

Figure 4C is not very intuitive.

The explanation of Figure 5 on page 14 and page 15 is very long. Please reduce.

The authors could consider citing recent work on phosphoproteomics-informed combinatorial treatments of PDAC (from the Jimenez lab).

Reviewer #3:

The manuscript by Hoefler et al. searched for synergistic combinations for the treatment of PDAC with gemcitabine (GEM). They performed a drug screen to find drug synergism and found ATRi to have synergistic effects in almost all PDAC cell lines tested. Next, they performed a deep mechanistic analysis, using a phosphoproteomic approach and kinobeads, to understand the mediators of drug synergism. While the synergism between GEM and DDR is already described, the study offers several novel aspects with potential therapeutic implications. The study is designed and described well. It starts with a broad screen, and ends with clinically-relevant combination therapy. Therefore, I find it suitable for publication in MSB after the implementation of several key additions.

1. Most in-vitro drug screens suffer from poor relevance to in-vivo situations, as they primarily feature cell proliferation rather than many other cancer features, such as invasiveness, migration, and metabolism. It is, therefore, not surprising that a DNA-damage-related kinase was picked up on the screen. While I value the potential importance of the in-vitro finding of ATRi-GEM synergism, the manuscript would highly benefit from an investigation of the in-vivo relevance of their findings. These validations can be in the form of tumor mouse models or examination of the relevance based on existing clinical proteomics and phosphoproteomics data.

2. ATRi was found to work synergistically with GEM. I wonder why other DDR inhibitors did not show a similar phenotype. The authors indicate that Chk1 inhibitors have a similar (but less robust) effect, while their screen shows that ATMi does not. Since ATM and ATR have a substantial overlap of their substrates, the authors should add ATMi phosphoproteomics data to further pinpoint relevant phosphosites to the GEM-ATRi synergism. I expect that many of the ATRi effects on GEM-treated cells would resemble those of ATMi.

3. The authors' hypothesis is that synergism between ATRi and GEM is caused by increased DNA damage due to a lack of DDR. However, in recent years, there have been indications of additional, non-canonical activities of ATR. The authors should address the potential involvement of these pathways, which may also partially explain the discrepancy between ATRi and other DDR inhibitors.

4. Line 228, what does the number 64,000 refer to? The authors indicate different numbers a few lines before.

5. Figure 5 should also present the sites in a comparison of GEM + ATRi vs. GEM alone (e.g., by a volcano plot). It is important to see the extent of ATRi reversal of the GEM effect.

Point to point response

The authors thank all reviewers for their time and thoughtful comments. In light of their advice, we have made substantial changes to the original manuscript which include the addition of new data, new data analysis, new figure panels and new text. The major additions are as follows:

- We have generated more phenotypic viability data. Specifically, we have verified drug synergy between GEM and four clinical ATR inhibitors in additional cell lines (PANC-1 and Suit2-07; Figure EV2A,B), and performed viability tests on GEM+ATRi Elimusertib in nine pancreatic PDOs (Figure 2E,F; Figure EV2E). Further, we have added new viability data testing other chemotherapies in combination with ATRi Elimusertib in cell line AsPC-1 (Figure EV2C,D).
- We have added a new phosphoproteomic data set on six additional PDAC cell lines treated with GEM plus ATRi Elimusertib to evaluate the robustness of the 36 proposed biomarker phosphorylation sites (Figure 5D,E). Additionally, based on this data, we verified ATR-dependent from ATR-independent phosphorylation events upon combination therapy (Appendix Fig. S1).
- We further mined public (phospho-)proteomic PDAC patient data (CPTAC project) for the proposed biomarkers to investigate their baseline levels in tumor vs. healthy tissue (Appendix Fig. S2).
- We have included new proteomic data on the temporal changes in protein expression upon GEM and ATRi Elimusertib (Appendix Fig. S3). The data was added to highlight potential involvement of non-canonical roles of ATR in drug synergy with GEM.
- We have made multiple changes to the manuscript text to include the additional data and analysis (see tracked changes in .docx file).

Reviewer #1:

In the paper "Gemcitabine and ATR inhibitors synergize to kill PDAC cells by blocking DNA damage response" Höfer et al. perform initially a phenotypic screen followed by a proteomic screen to test synergy of the DNA-damage agent Gemcitabine (GEM) with kinase inhibitors in 13 human pancreatic adenocarcinoma cell lines. They found that generally ATR inhibitors (ATRi) synergize with GEM across the majority of cell lines. Furthermore, using dose-dependent phosphoproteomics on cells that were treated with GEM alone compared to the combination treatment with (ATRi), the authors could provide hypotheses for the synergistic effect and suggest possible phosphorylation sites that might be critical for the synergy.

Overall, I think this is an elegant, comprehensive, and well-performed study to test drug synergies across multiple cell lines and many drugs. While the overall outcome on the synergies is not entirely novel as GEM and ATRi synergies have been reported before, the novelty of the study in my opinion lies in providing (1) a general solution for in vitro testing and prioritizing drug synergies, which can be expanded to other targets and (2) a data resource that can be mind

to derive mechanistic hypothesis for more targeted follow up studies.

There are a few additional experiments and comments that the authors should address in the current version before publication.

Major comments

- While the panel of 4 ATRi in combination with GEM are only tested in AsPC-1 cells, the question remains whether the inhibition of ATR has a broad synergistic effect with GEM across the other cell lines tested or whether the broad synergistic effect is specific for Elimuserib?

We have added new data testing the viability of PANC-1 and Suit2-07 cells against the combination of GEM and four clinical ATRis (including Elimuserib). Synergy was observed in both cell lines (see plots below) thus confirming that the synergistic effect is quite broad. This data has been added to the manuscript as new Figure EV2 (panels A and B) and as the following text of the results section: *'The same experimental setup was repeated in cell lines PANC-1 and Suit2-07, where all four drug combinations demonstrated superior efficacy compared to GEM monotherapy (Fig. EV2A,B; Dataset EV6).'* (lines 202-204).

Moreover, we have expanded the tested PDAC models to include nine pancreatic cancer PDOs (see response to reviewer #3 below).

- Based on the DecryptM phosphoproteomics approach the authors can pinpoint specific phosphorylation sites that might be indicative of the synergistic effect of GEM and ATRi, among these are sites that have not previously been linked to DNA damage. While it would be very exciting to test these sites functionally to validate their effect on DNA damage, these experiments would be beyond the scope of this study. However, the authors suggest these sites could serve as biomarkers for drug response for the combination of GEM and ATRi. One important aspect of a drug response marker is that these markers show a consistent effect that is not specific for a certain drug or cell lines. While the authors provide data that these phosphorylation site changes are consistent across different ATRi inhibitors in one cell line, it would be important to assess whether these phosphorylation sites are also consistently responding across the different cell lines. Therefore, I would suggest an experiment to test the consistency of the phosphorylation sites changes across the panel of cell lines for the GEM/Elimuserib combination. This would not only strengthen the claim for certain

phosphosites as response markers, which I believe represents a novelty of the study, but would in my opinion also be within the scope of this study.

We have performed additional phosphoproteomics experiments with GEM+Elimusertib in six further PDAC cell lines from the panel. Thirty two of the 36 pSQ/pTQ sites were also identified in these cell lines and all but one showed a combined effect GEM monotherapy (see plot below, left). Importantly, as seen in AsPC-1 cells, the CHEK1-pS468 site was most strongly affected (see plot below, right). This confirms the consistency of the proposed biomarkers in general, and CHEK1-pS468 in particular. We included this data as new Figure 5 (panels D and E) and added the following text to the results section:

‘To assess the robustness of the proposed biomarkers, we examined their regulation in six additional PDAC cell lines (PANC-1, MiaPaCa-2, PaTu-8988-T, PSN-1, Dan-G, and Capan-1) treated with 1 μM GEM, 1 μM GEM plus 1 μM Elimusertib, or vehicle. Of the 36 pSQ/pTQ sites, 32 were quantified, with the majority showing > 2-fold reduction upon combination treatment compared to GEM alone (Fig. 5E; Dataset EV12). Notably, strong inhibition was observed for the known ATR substrates CHEK1-pS317 and FANCD2-pS319, as well as the novel substrate candidate LRCC42-pS385 (log₂ fold changes < -2 in at least three cell lines). Consistent with the AsPC-1 results, CHEK1-pS468 was most strongly inhibited, with log₂ fold changes as low as -4.5 and -4.2 in Capan-1 and Dan-G cells (adj. p-values < 8e⁻³ and < 4e⁻³, respectively; Fig. 5F; Dataset EV11). [...] These results highlight the consistent regulation of the proposed pSQ/pTQ sites in response to GEM and ATRi, underscoring their potential as biomarkers for treatment response.’ (lines 355-368).

Minor comments

- It would be beneficial for the reader if some additional description for the phenotypic drug screen would be provided in the first section of the manuscript. Currently, it is not apparent from the text what the phenotypic readout for the drug screen actually is, only by looking at the figure readers will know that there is a cell viability readout.

We have added additional information regarding the drug screen viability readout to the beginning of the results section as follows:

- *‘To systematically identify established drugs that possibly synergize with GEM, we performed a phenotypic cell viability screen with 146 targeted agents alone as well as in combination with GEM [...]’ (lines 103-105)*
- *‘Cell viability was assessed by measuring ATP levels to indicate metabolic activity.’ (lines 113-114)*

Reviewer #2

This is an important and well-written piece of work. I have only few comments:

MAJOR CONCERNS

Gemcitabine is not really the mainstay of systemic treatment of PDAC anymore. The authors should provide a stronger rationale why gemcitabine was the backbone for testing combinations.

We acknowledge that the past years have seen a shift towards other drugs and combinations. We had two main reasons for choosing GEM: First, the author's laboratory has a very active research program on kinase inhibitors in cancer. This is why we did not prioritize the combination of GEM and Paclitaxel for the present work. In addition, we were intrigued by the large number of clinical trials that combined GEM with kinase inhibitors over the past 10 years (n=24) which we took as a rationale to start our project (several years ago) by screening a substantial number of (clinical) kinase inhibitors. Second, we aimed at generating a better understanding of the molecular mechanisms underlying the drug combination. This would have been very difficult if not impossible to do if we had chosen one of the more complex current treatment regimen such as (m)FOLFIRINOX.

We have changed the text in the introduction to bring this out more clearly:

- *'For many years, the nucleoside analogue Gemcitabine (GEM) has been a frontline drug in PDAC, inducing DNA damage and replication stress response in dividing cells by disrupting DNA synthesis (Burris et al, 1997; Park et al., 2021; Plunkett et al, 1995). In current clinical practice, GEM is often administered in combination with albumin-bound Paclitaxel particles (nab-Paclitaxel) as the combination substantially improves therapeutic efficacy (Kang et al, 2018).'* (lines 48-53)
- *'Currently, at least 30 kinase inhibitors are under clinical investigation for PDAC (phase II or III), either as single agents or in combination with chemotherapy, most commonly GEM (Fang et al., 2023).'* (lines 69-70)

Related to that; can the authors demonstrate that the synergy is specific to gemcitabine and not the other commonly used cytotoxics (paclitaxel, 5-FU, oxaliplatin, irinotecan/SN-38)?

We have added further data testing proliferation in response to ATR inhibition in combination with 5-FU, Oxaliplatin, Cisplatin, or Paclitaxel in AsPC-1 cells. The data showed (see figure below) that ATRi synergized with 5-FU, Oxaliplatin, Cisplatin (all DNA-damaging agents) but not Paclitaxel (microtubule destabilizer). We thus conclude that other DNA-damaging chemodrugs might also benefit from the addition of ATRi. This new data has been added as Figure EV2C,D and we added the following text to the results section:

'To determine whether drug synergy is specific to GEM only, Elimusertib was also evaluated in combination with other commonly used chemotherapeutic agents in AsPC-1 cells. Elimusertib sensitized AsPC-1 cells to the three DNA-damaging agents Oxaliplatin, Cisplatin, and 5-fluorouracil, but not to Paclitaxel (Fig. EV2C,D; Dataset EV7). This suggests that drug synergy with ATRi is not limited to GEM but also applies to other DNA-damaging agents, further highlighting the critical role of disrupted DNA repair in the synergy mechanism.' (lines 208-214)

HPDE cells are not normal or healthy. The authors can remove the caveat at the bottom of page 9 but I would also advise to test the combination in organoids. In this system, truly healthy cells can be compared to tumour-derived organoids. This would also add to the translational potential (in lieu of mouse experiments that I do not recommend).

We have removed the following sentence from the manuscript: *'Of note, immortalized duct cells were also weakly chemosensitized by Elimusertib ($\Sigma AUC_{\Delta} = 15\%$), indicating that toxicity may be observed in patients'*.

Moreover, we have expanded the tested PDAC models to include nine pancreatic cancer PDOs (see response to reviewer #3 below).

MINOR COMMENTS

I feel that a lot of what is now in the Extended Data could actually be shown in the main figures as well. I do not think it would add too much bulk.

As part of the revision process, we made changes to the main figures to accommodate the new data (e. g. the PDOs in main Figure 2 and additional phosphoproteomic data in Figure 5) which has increased the size of some figures. Therefore, for the sake of clarity, we have not promoted any of the supplementary material to the main figures.

What type of phosphoproteomics was performed is not immediately clear in the Results or Figure Legend. Please add.

We have added some information to the text to make clear that the phosphoproteomic data was acquired by mass spectrometry:

- Results: *'To investigate if and how the above ATR inhibitors engage their targets in cells, we followed the decryptM approach (Zecha et al., 2023) and performed mass spectrometry-based phosphoproteomics on GEM-induced DNA-damaged and subsequently ATRi-treated cells (Figure EV3A).'* (lines 247-250)

- Figure 4A legend: *'Potencies (pEC₅₀) and curve fold changes (log₂) from phosphoproteomic decryptM experiments with Elimusertib and Gartisertib (top) and Berzosertib and Ceralasertib (bottom) in DNA-damaged cells acquired by LC-MS/MS.'*

Figure 2 has an L-shaped line around the figure.

We have removed the frame around Figure 2.

Figure 4C is not very intuitive.

We now explain the plot in more detail. The intent of the plot is to compare the effects of the four drugs on the phospho-proteome. The x-axis represents the potency of the drugs (pEC₅₀). The y-axis shows the density, reflecting the distribution of these potency values across the drug-responsive phosphorylation sites (SQ/TQ only, indicating DNA damage signaling). It shows that the majority of pSQ/pTQ sites are >10-times more potently affected by Elimusertib and Gartisertib (pEC₅₀ ~8 = 10 nM) than Berzosertib and Ceralasertib (pEC₅₀ ~7 = 100 nM). We have added the following text to clarify this:

- Results: *'The dose-dependent decryptM approach revealed that the four drugs inhibited cellular DNA damage signaling (via pSQ/pTQ-sites) at different potencies. When summarizing EC₅₀ values across all drug-regulated pSQ/pTQ-sites, Elimusertib and Gartisertib were 10-20-fold more potent in cells (median EC₅₀ of 10 nM and 21 nM, respectively) than Ceralasertib and Berzosertib (median EC₅₀ of 154 nM and 188 nM, respectively; Figure 4C).'* (lines 271-276)
- Figure 4C legend: *'Density plot summarizing the drug potencies (pEC₅₀) by which the four ATRi regulate the phosphorylation of pSQ/pTQ-peptides in DNA-damaged cells. The number of pSQ/pTQ-peptides regulated per drug is given in the legend.'*

Original Figure 4C (not modified):

The explanation of Figure 5 on page 14 and page 15 is very long. Please reduce.

We have substantially reduced the original Figure legend of Figure 5.

The authors could consider citing recent work on phosphoproteomics-informed combinatorial treatments of PDAC (from the Jimenez lab).

We have added the following text to the discussion section and cited Vallés-Martí et al. 2023 from the Jimenez lab (Cell Reports, Volume 42, Issue 6, 112581):

‘In this context, baseline (phospho-)proteomes and kinase activity landscapes of the cell panel could further help prioritize promising drug combinations, as previously demonstrated by Vallés-Martí et al. (2023).’ (lines 469-472)

Reviewer #3

The manuscript by Hoefler et al. searched for synergistic combinations for the treatment of PDAC with gemcitabine (GEM). They performed a drug screen to find drug synergism and found ATRi to have synergistic effects in almost all PDAC cell lines tested. Next, they performed a deep mechanistic analysis, using a phosphoproteomic approach and kinobeads, to understand the mediators of drug synergism. While the synergism between GEM and DDR is already described, the study offers several novel aspects with potential therapeutic implications. The study is designed and described well. It starts with a broad screen, and ends with clinically-relevant combination therapy. Therefore, I find it suitable for publication in MSB after the implementation of several key additions.

1. Most in-vitro drug screens suffer from poor relevance to in-vivo situations, as they primarily feature cell proliferation rather than many other cancer features, such as invasiveness, migration, and metabolism. It is, therefore, not surprising that a DNA-damage-related kinase was picked up on the screen. While I value the potential importance of the in-vitro finding of ATRi-GEM synergism, the manuscript would highly benefit from an investigation of the in-vivo relevance of their findings. These validations can be in the form of tumor mouse models or examination of the relevance based on existing clinical proteomics and phosphoproteomics data.

The authors acknowledge that testing *in-vivo* models would add value but the authors think mouse experiments are outside of the scope of the current study. We have talked to colleagues who have PDAC mouse models but the regulatory steps required to perform such experiments would take many months leading to unreasonable delays. Instead, we have performed additional viability experiments of GEM+Elimusertib in nine pancreatic PDOs, and observed drug synergy in five (see below). We have included the new data in Figure 2 (panels C and D) and added the following text to the results section:

'To evaluate the synergistic effects in more advanced disease models, nine PDAC patient-derived organoids (PDOs) were tested for their response to GEM combined with ATRi Elimusertib. Sensitivity to the individual drugs as monotherapies varied among the PDOs, ranging from 3 nM to 285 nM for GEM and from 43 nM to 290 nM for Elimusertib (Dataset EV8). Upon the addition of sub-EC50 doses of ATRi (30 nM or lower), five out of nine PDOs were significantly sensitized to GEM in at least one treatment condition (> 2-fold reduction in EC50 and adj. p-value < 0.05 compared to GEM monotherapy; Fig. X). The remaining four PDOs remained mostly unaffected. These results suggest that this regimen may be efficacious only in specific biological contexts, underscoring the importance of identifying predictive response markers for GEM and ATRi.' (lines 214-223)

Figure 2 (panels C and D):

Figure EV2 (panel E):

In addition, we have mined phosphoproteomic data collected from PDAC patients as part of the CPTAC project for pSQ/pTQ sites associated with drug synergy in our work (<https://cprosite.ccr.cancer.gov/#/>). Of the 36 pSQ/pTQ sites discussed in our manuscript, 8 were found in patients and all but one showed higher phosphorylation levels in PDAC tumors compared to adjacent normal tissue (see plot A below). We also compared the protein expression levels of ATR and CHEK1 in these CPTAC cohort and found that PDAC tumor cells express statistically significantly higher levels of CHEK1 than adjacent normal cells (see plot B below). We have added this analysis to the supplementary material (Appendix 1) and added the following text to the manuscript:

'Additionally, we have mined phosphoproteomic PDAC patient data as part of the CPTAC project, focusing on the 36 pSQ/pTQ sites linked to drug synergy. Eight of these sites were detected in patients, with all but one showing elevated phosphorylation levels in PDAC tumors compared to adjacent normal tissue (Appendix Fig. S2A). Moreover, protein expression analysis indicated that although ATR is not differentially expressed in PDAC, tumor cells exhibit significantly higher CHEK1 levels than normal tissue (Appendix Fig. S2B). While no formal

proof, these data imply that there may be increased DNA damage response activity in tumor cells, which may be repressed by ATR inhibition.’ (lines 369-376)

2. ATRi was found to work synergistically with GEM. I wonder why other DDR inhibitors did not show a similar phenotype. The authors indicate that Chek1 inhibitors have a similar (but less robust) effect, while their screen shows that ATMi does not. Since ATM and ATR have a substantial overlap of their substrates, the authors should add ATMi phosphoproteomics data to further pinpoint relevant phosphosites to the GEM-ATRi synergism. I expect that many of the ATRi effects on GEM-treated cells would resemble those of ATMi.

GEM is reported to primarily induce replication stress that is countered by the action of ATR while ATM is important for DNA double strand break sensing and repair. We acknowledge, that ATR and ATM have overlapping substrates, but our data and analysis clearly pinpoints ATR as the relevant kinase in the context of GEM. First, kinobead analysis (Figure 3B) showed that none of the four ATR inhibitors binds ATM. Second, as the reviewer points out, ATM inhibition did not elicit a phenotypic effect (Figure 1B, Figure EV1C). Third, phosphoproteomic profiling showed that while many ATM substrates were up-regulated by GEM, their phosphorylation was unaffected by ATR inhibition (Figure 5D). Fourth, in response to reviewer #1 above, we generated phospho-proteomes for six additional cell lines in response to ATR inhibition. We analysed this data for well-described ATR and ATM substrates (see plot A and B below, respectively) and found that the ATM sites were not affected by GEM/Elimusertib. We have added this data as new Appendix Figure S1 and highlighted it in the results section as follows: *‘Importantly, known substrates of the related kinase ATM, such as ATM-pS1981, H2AFX-pS139, BRCA1-pS1524, and TRIM28-pS824 (all pSQ/pTQ), remained unaffected by ATRi across all cell lines, further suggesting ATR-specific regulation (Appendix Fig. S1).’ (lines 364-366)*

We have also added the following text to the discussion section to point out that looking at ATM would be a useful future extension of the work when investigating drug synergy for drugs that induce other types of DNA damage and which are repaired under the participation of ATM or other kinases:

‘In future work, phosphoproteomic studies using ATM inhibitors on GEM-treated cells could complement the current work as GEM these may identify further molecular mechanisms that are based on DNA damage, helping to pinpoint overlapping and distinct substrates that contribute to GEM-ATRi synergy.’ (lines 478-481)

3. The authors' hypothesis is that synergism between ATRi and GEM is caused by increased DNA damage due to a lack of DDR. However, in recent years, there have been indications of additional, non-canonical activities of ATR. The authors should address the potential involvement of these pathways, which may also partially explain the discrepancy between ATRi and other DDR inhibitors.

We have performed additional drug-perturbation experiments in AsPC-1 cells and measured protein expression changes over time upon GEM or Elimusertib. We found that several nucleotide metabolism enzymes were regulated in opposite directions by GEM and Elimusertib (see below). Nucleotide metabolism is one of the described non-canonical functions of ATR and GEM is also described to adversely affect this process. Among the regulated proteins is RRM2 (but not RRM1), one of the key enzymes in the pathway and described to be regulated by ATR as well as GEM. TK1 (thymidine kinase) is also among the regulated proteins, further strengthening this connection. Therefore, it is well possible that this function of ATR adds to the observed drug synergy. We have added the new data as Appendix Figure S3 and included the following text in the manuscript: *Moreover, while our study identified disrupted DNA damage response as the primary mechanism of drug synergy, there is growing evidence of non-canonical*

activities of ATR that may contribute to these combination effects (da Costa et al, 2023). Notably, prolonged exposure of PDAC cells to GEM and ATRi resulted in differential expression of enzymes involved in nucleotide metabolism, specifically RRM2 (but not RRM1) and TK1 (Appendix Fig. S3; Dataset EV12). Given that both ATR and GEM are known to impact cellular nucleotide pools, this dysregulation likely represents an additional mechanism driving GEM-ATRi synergy (da Costa et al., 2023; Huang et al, 1991). These findings prompt further investigation into such non-canonical ATR pathways, which may explain the superior synergy between GEM and ATRi compared to other DNA damage response inhibitors.’ (lines 481-490)

4. Line 228, what does the number 64,000 refer to? The authors indicate different numbers a few lines before.

This is the (rounded) total number of dose-response curves generated across all four decryptM experiments. It was meant to give readers an idea of the scale of the phosphoproteomic data that underlies our analysis. The accurate number can be extracted from Dataset EV7 (count of rows per experiment). To clarify:

- decryptM with GEM+Elimusertib: 15,137
- decryptM with GEM+Berzosertib: 17,314
- decryptM with GEM+Ceralasertib: 16,767
- decryptM with GEM+Gartisertib: 15,014
- **Total: 64,232 dose-response curves**

The number of 20,784 phosphorylation sites mentioned a few lines before, refers to the non-redundant number of all phosphorylation-sites that are contained in the 64,000. The numbers are not the same as many phosphorylation-sites were identified in multiple experiments. We clarified this in the text as follows:

‘Across all experiments, 25,537 non-redundant phosphorylated peptides were quantified, of which 20,784 had phosphorylation sites (p-sites) that could be localized with a probability of > 0.75 (15,014 to 17,314 p-sites per experiment; Dataset EV7). Only the latter were used for all subsequent analyses. The underlying ~64,000 p-site dose-response profiles were statistically evaluated and categorized into significant up- and down-regulated curves by CurveCurator (Bayer et al, 2023) (see methods for details).’ (lines 252-257)

5. Figure 5 should also present the sites in a comparison of GEM + ATRi vs. GEM alone (e.g., by a volcano plot). It is important to see the extent of ATRi reversal of the GEM effect.

This data was already included in the original submission as Figure EV4B and we have added a new lollipop plot to main Figure 5 (panel C, see below) that shows the extent of reversal as requested. Alongside, we have added the following text to the manuscript:

‘The most strongly affected p-site was CHEK1-pS468 (pSQ/pTQ). Phosphorylation of this site was increased > 18-fold upon GEM treatment (log₂ fold change of 4.2, adjusted p-value < 0.002) and reversed up to 14-fold by three of the four ATRi (not quantified for Gartisertib; log₂ fold change upon ATRi between -3.8 and -2.9; Fig. 5B,C).’ (lines 304-307)

26th Nov 2024

Manuscript Number: MSB-2024-12345R

Title: Gemcitabine and ATR inhibitors synergize to kill PDAC cells by blocking DNA damage response

Dear Dr. Küster,

Thank you for the submission of your revised manuscript to Molecular Systems Biology. We have now received the enclosed reports from the referees that were asked to re-assess it. As you will see the reviewers are now globally supportive and I am pleased to inform you that we will be able to accept your manuscript pending the following final amendments:

1) Please update the Data availability section to be formatted according to the example below:

"The datasets and computer code produced in this study are available in the following databases:

- Chip-Seq data: Gene Expression Omnibus GSE46748 (<https://www.ncbi.nlm.nih.gov/geo/query/acc.cgi?acc=GSE46748>)

- Modeling computer scripts: GitHub (<https://github.com/SysBioChalmers/GECKO/releases/tag/v1.0>)

- [data type]: [full name of the resource] [accession number/identifier] ([doi or URL or identifiers.org/DATABASE:ACCESSION])"

2) Data availability: Please also ensure that the proteomics dataset is now publicly available and remove the reviewer access information from the Data Availability statement. A URL as noted above should also be included.

3) Please rename separate the "Disclosure and competing interests statement" from the Author Contributions section. We updated our journal's competing interests policy in January 2022 and request authors to consider both actual and perceived competing interests. Please review the policy <https://www.embopress.org/competing-interests> and update your competing interests if necessary.

4) Author contributions: Please remove it from the manuscript and specify author contributions in our submission system. CRediT has replaced the traditional author contributions section because it offers a systematic machine-readable author contributions format that allows for more effective research assessment. You are encouraged to use the free text boxes beneath each contributing author's name to add specific details on the author's contribution. More information is available in our guide to authors:

<https://www.embopress.org/page/journal/17574684/authorguide#authorshipguidelines>

5) Our journal encourages inclusion of *data citations in the reference list* to directly cite datasets that were re-used and obtained from public databases. Data citations in the article text are distinct from normal bibliographical citations and should directly link to the database records from which the data can be accessed. In the main text, data citations are formatted as follows: "Data ref: Smith et al, 2001" or "Data ref: NCBI Sequence Read Archive PRJNA342805, 2017". In the Reference list, data citations must be labeled with "[DATASET]". A data reference must provide the database name, accession number/identifiers and a resolvable link to the landing page from which the data can be accessed at the end of the reference. Further instructions are available at .

6) Data citations: Data citations and references should include links to specific accession codes used in the study, not to a general database. In studies that make use of many (i.e. more than approx. 20) pre-existing datasets as in this case, we realize it is not practical or feasible to cite them individually in the main manuscript. In this case, it is therefore acceptable to provide a separate data reference list in the form of an Expanded View Table which should be called out from the Methods section. Further instructions are available at .

7) In the Methods, please take care of the following:

- The Materials and Methods section should be renamed to "Methods".

- Studies with human research participants: The use of human samples requires information on the authority granting ethics approval (e.g. IRB) and informed consent. If the need for approval is waived, please cite the reason (e.g. non-human subject research because the samples used were de-identified/coded with no identifying information) and legislation in the relevant methods section.

- Please ensure that a statement on whether or not blinding was done is included in the Methods even if no blinding was done. Please also be sure to update the Author Checklist with this information and where it can be found in the manuscript.

8) Please upload the Reagents and Tools table as a .docx file. You can find the template in our author guidelines:

9) Please place individual sections of the manuscript in the following order: Title page with complete author information - Abstract & Keywords - Introduction - Results - Discussion - Methods - Data Availability - Acknowledgements - Disclosure and Competing Interests Statement - References - Figure Legends - Expanded View Figure Legends.

10) For the figures and figure legends, please take care of the following:

- Please make sure to update the callouts of all figures in the main manuscript text. Currently figure callouts are missing for Supplementary Table S1, which should be renamed to Appendix Table S1.

- Please note that the box plots need to be defined in terms of minima, maxima, centre, bounds of box and whiskers, and percentile in the legends of figures S2 B.

- Please note that information related to n is missing in the legends of figures 5F; EV1 B; S1, S2 A, B.

- Please note that n=2 in figure EV2 B.

- Although 'n' is provided, please describe the nature of entity for 'n' in the legends of figures 5A, EV2 B.

- Please note that the error bars are not defined in the legends of figures S2 A.

- Please note that the measure of center for the error bars needs to be defined in the legends of figures 2A-D; EV1 B; EV2 A-E; S1.
 - Please note that the legends of figure 3 A-C is mislabeled as figure E-G in the manuscript. This needs to be rectified.
 - Please note that the legends of figure 5 A-F is mislabeled as figure K-P in the manuscript. This needs to be rectified.
 - Please note that the legends of figure EV1 A-C is mislabeled as figure Q-S in the manuscript. This needs to be rectified.
 - Please note that the legends of figure EV3 A-B is mislabeled as figure F-G in the manuscript. This needs to be rectified.
 - Please note that the legends of figure EV4 A-C is mislabeled as figure H-J in the manuscript. This needs to be rectified.
 - Please note that the legends of figure EV5 A-D is mislabeled as figure K-N in the manuscript. This needs to be rectified.
 - Please note that the exact p values are not provided in the legends of figures 2B, D; EV2D.
 - Please indicate the statistical test used for data analysis in the legend of figure S2 B.
- 11) Each Expanded View Dataset will need its legend removed from the manuscript and added to the corresponding EV Dataset file in a separate tab.
 - 12) Please add page numbers to the Table of Contents in the Appendix file.
 - 13) Please check your synopsis text and image before submission with your revised manuscript. Please be aware that in the proof stage minor corrections only are allowed (e.g., typos).
 - 14) As part of the EMBO Publications transparent editorial process initiative (see our policy here: https://www.embopress.org/transparent-process#Review_Process), Molecular Systems Biology will publish online a Peer Review File (PRF) to accompany accepted manuscripts. This file will be published in conjunction with your paper and will include the anonymous referee reports, your point-by-point response and all pertinent correspondence relating to the manuscript. Let us know whether you agree with the publication of the PRF and as here, if you want to remove or not any figures from it prior to publication. Please note that the Authors checklist will be published at the end of the PRF.
 - 15) Please provide a point-by-point letter INCLUDING my comments and your detailed responses (as Word file).

I look forward to reading a new revised version of your manuscript as soon as possible.

Yours sincerely,

Poonam Bheda, PhD
Scientific Editor
Molecular Systems Biology

Reviewer #1:

In the revised version of the manuscript the authors have sufficiently addressed my comments/concerns raised in the initial submission, so that the manuscript is in my opinion acceptable for publication.

Reviewer #2:

The authors have addressed my concerns and I have no other comments. Job well done.

Reviewer #3:

The authors answered all of my requests. I therefore find the manuscript suitable for publication.

Manuscript Number: MSB-2024-12345R

Title: Gemcitabine and ATR inhibitors synergize to kill PDAC cells by blocking DNA damage response

Dear Dr. Küster,

Thank you for the submission of your revised manuscript to Molecular Systems Biology. We have now received the enclosed reports from the referees that were asked to re-assess it. As you will see the reviewers are now globally supportive and I am pleased to inform you that we will be able to accept your manuscript pending the following final amendments:

1) Please update the Data availability section to be formatted according to the example below:

"The datasets and computer code produced in this study are available in the following databases:

- Chip-Seq data: Gene Expression Omnibus GSE46748

(<https://www.ncbi.nlm.nih.gov/geo/query/acc.cgi?acc=GSE46748>)

- Modeling computer scripts: GitHub (<https://github.com/SysBioChalmers/GECKO/releases/tag/v1.0>)

- [data type]: [full name of the resource] [accession number/identifier] ([doi or URL or identifiers.org/DATABASE:ACCESSION])"

Done.

2) Data availability: Please also ensure that the proteomics dataset is now publicly available and remove the reviewer access information from the Data Availability statement. A URL as noted above should also be included.

Done.

3) Please rename separate the "Disclosure and competing interests statement" from the Author Contributions section. We updated our journal's competing interests policy in January 2022 and request authors to consider both actual and perceived competing interests. Please review the policy <https://www.embopress.org/competing-interests> and update your competing interests if necessary.

Done.

4) Author contributions: Please remove it from the manuscript and specify author contributions in our submission system. CRediT has replaced the traditional author contributions section because it offers a systematic machine-readable author contributions format that allows for more effective research assessment. You are encouraged to use the free text boxes beneath each contributing author's name to add specific details on the author's contribution. More information is available in our guide to authors:

<https://www.embopress.org/page/journal/17574684/authorguide#authorshipguidelines>

Done.

5) Our journal encourages inclusion of *data citations in the reference list* to directly cite datasets that were re-used and obtained from public databases. Data citations in the article text are distinct from normal bibliographical citations and should directly link to the database records from which the data can be accessed. In the main text, data citations are formatted as follows: "Data ref: Smith et al, 2001" or "Data ref: NCBI Sequence Read Archive PRJNA342805, 2017". In the Reference list, data citations must be labeled with "[DATASET]". A data reference must provide the database name, accession number/identifiers and a resolvable link to the landing page from which the data can be accessed at the end of the reference. Further instructions are available at <https://www.embopress.org/page/journal/17574684/authorguide#referencesformat>.

Done. We have included data citations for the two external datasets used, and formatted them according to the author guidelines.

6) Data citations: Data citations and references should include links to specific accession codes used in the study, not to a general database. In studies that make use of many (i.e. more than approx. 20) pre-existing datasets as in this case, we realize it is not practical or feasible to cite them individually in the main manuscript. In this case, it is therefore acceptable to provide a separate data reference list in the form of an Expanded View Table which should be called out from the Methods section. Further instructions are available at <https://www.embopress.org/page/journal/17574684/authorguide#referencesformat>.

See response to 5). We have cited the two datasets individually in the main text and included the specific accession codes in the references.

7) In the Methods, please take care of the following:

- The Materials and Methods section should be renamed to "Methods".

Done.

- Studies with human research participants: The use of human samples requires information on the authority granting ethics approval (e.g. IRB) and informed consent. If the need for approval is waived, please cite the reason (e.g. non-human subject research because the samples used were de-identified/coded with no identifying information) and legislation in the relevant methods section.

The following information on the ethics approval and informed consent was included in the Methods section (in the end of "Cell lines and patient-derived organoids"):

"Patient material used for this study was approved by the local ethics committees (Munich: project 207/15, Essen: 22-10622-BO), and written informed consent was obtained from the patient prior to the investigation."

Also, we have updated the Author Checklist and added a reference to the Methods section.

- Please ensure that a statement on whether or not blinding was done is included in the Methods even if no blinding was done. Please also be sure to update the Author Checklist with this information and where it can be found in the manuscript.

We have included the information in the Methods section (in the end of "Cell lines and patient-derived organoids") and updated the Author Checklist.

8) Please upload the Reagents and Tools table as a .docx file. You can find the template in our author guidelines: <https://www.embopress.org/page/journal/14693178/authorguide#structuredmethods>.

Done.

9) Please place individual sections of the manuscript in the following order: Title page with complete author information - Abstract & Keywords - Introduction - Results - Discussion - Methods - Data Availability - Acknowledgements - Disclosure and Competing Interests Statement - References - Figure Legends - Expanded View Figure Legends.

Done.

10) For the figures and figure legends, please take care of the following:

- Please make sure to update the callouts of all figures in the main manuscript text. Currently figure callouts are missing for Supplementary Table S1, which should be renamed to Appendix Table S1.

Done.

- Please note that the box plots need to be defined in terms of minima, maxima, centre, bounds of box and whiskers, and percentile in the legends of figures S2 B.

Done.

- Please note that information related to n is missing in the legends of figures 5F; EV1 B; S1, S2 A, B.

Done.

- Please note that n=2 in figure EV2 B.

The data in Figure EV2B (Suit2-07 cells) were obtained from a single experiment (n = 1). This information has been added at the end of the legend for clarity.

- Although 'n' is provided, please describe the nature of entity for 'n' in the legends of figures 5A, EV2 B.

We have added the following information for Figure 5A: *"n = 4 independent experiments"* and for Figure EV2 B: *"Data shown in B represent a single experiment (n = 1)."* (see response above).

- Please note that the error bars are not defined in the legends of figures S2 A.

Done.

- Please note that the measure of center for the error bars needs to be defined in the legends of figures 2A-D; EV1 B; EV2 A-E; S1.

Done.

- Please note that the legends of figure 3 A-C is mislabeled as figure E-G in the manuscript. This needs to be rectified.

Done.

- Please note that the legends of figure 5 A-F is mislabeled as figure K-P in the manuscript. This needs to be rectified.

Done.

- Please note that the legends of figure EV1 A-C is mislabeled as figure Q-S in the manuscript. This needs to be rectified.

Done.

- Please note that the legends of figure EV3 A-B is mislabeled as figure F-G in the manuscript. This needs to be rectified.

Done.

- Please note that the legends of figure EV4 A-C is mislabeled as figure H-J in the manuscript. This needs to be rectified.

Done.

- Please note that the legends of figure EV5 A-D is mislabeled as figure K-N in the manuscript. This needs to be rectified.

Done.

- Please note that the exact p values are not provided in the legends of figures 2B, D; EV2D.

Exact p-values for the most significant drug combinations were annotated directly in the plots of Figures 2B, 2D, and EV2D, and all others were referenced in their respective EV Tables.

- Please indicate the statistical test used for data analysis in the legend of figure S2 B.

Done.

11) Each Expanded View Dataset will need its legend removed from the manuscript and added to the corresponding EV Dataset file in a separate tab.

Done.

12) Please add page numbers to the Table of Contents in the Appendix file.

Done.

13) Please check your synopsis text and image before submission with your revised manuscript. Please be aware that in the proof stage minor corrections only are allowed (e.g., typos).

Done.

14) As part of the EMBO Publications transparent editorial process initiative (see our policy here: https://www.embopress.org/transparent-process#Review_Process), Molecular Systems Biology will publish online a Peer Review File (PRF) to accompany accepted manuscripts. This file will be published in conjunction with your paper and will include the anonymous referee reports, your point-by-point response and all pertinent correspondence relating to the manuscript. Let us know whether you agree with the publication of the PRF and as here, if you want to remove or not any figures from it prior to publication. Please note that the Authors checklist will be published at the end of the PRF.

We agree with the publication of the PRF, and we do not want to remove any figures from it prior to publication.

15) Please provide a point-by-point letter INCLUDING my comments and your detailed responses (as Word file).

Done.

I look forward to reading a new revised version of your manuscript as soon as possible.

Yours sincerely,

Poonam Bheda, PhD

Scientific Editor

Molecular Systems Biology

Reviewer #1:

In the revised version of the manuscript the authors have sufficiently addressed my comments/concerns raised in the initial submission, so that the manuscript is in my opinion acceptable for publication.

Reviewer #2:

The authors have addressed my concerns and I have no other comments. Job well done.

Reviewer #3:

The authors answered all of my requests. I therefore find the manuscript suitable for publication.

3rd Jan 2025

Manuscript number: MSB-2024-12345RR

Title: Gemcitabine and ATR inhibitors synergize to kill PDAC cells by blocking DNA damage response

Dear Dr. Küster,

Congratulations on an excellent manuscript, I am pleased to inform you that your manuscript has been accepted for publication in Molecular Systems Biology. Thank you for your comprehensive response to referee concerns. It has been a pleasure to work with you to get this to the acceptance stage.

Yours sincerely,

Poonam Bheda, PhD
Scientific Editor
Molecular Systems Biology
